



# Can we detect regional methane anomalies? A comparison between three observing systems.

Cindy Cressot[1], Isabelle Pison[1], Peter J. Rayner[2], Philippe Bousquet[1], Audrey Fortems-Cheiney[1], and Frédéric Chevallier[1]

[1]Laboratoire des Sciences du Climat et de l'Environnement, CEA/CNRS/UVSQ, Gif-sur-Yvette, France.
[2]School of Earth Sciences, University of Melbourne, Melbourne, Australia

*Correspondence to:* I. Pison (isabelle.pison@lsce.ipsl.fr)

**Abstract.** A Bayesian inversion system is used to evaluate the capability of the current global surface network and the space-borne GOSAT and IASI instruments to quantify surface flux anomalies of methane at various spatial (global, semi-hemispheric and regional) and time (seasonal, yearly, 3-yearly) scales. The evaluation is based on a signal-to-noise ratio analysis, the signal being the methane fluxes inferred from the surface-based inversion from 2000 to 2011 and the noise being computed

from the Bayesian equation for each of the three inversions using either surface or satellite data. At the global and semi-hemispheric scales, all observing systems properly detect flux anomalies at all the tested time-scales. At the regional scale, seasonal flux signals are properly detected by all observing systems, but year-to-year changes and longer-term trends are only poorly detected. Moreover, reliably detected regions depend on the reference surface-based inversion used as a signal. Indeed, tropical flux inter-annual variability, for instance, can be attributed mostly to Africa in the reference inversion or spread between

tropical regions and China. Our results show that inter-annual analyses of methane emissions inferred by atmospheric inversions should always include an uncertainty assessment and that the attribution of the atmospheric methane increase since 2007 to a particular region still needs more attention i.e. gathering more observations for the future and using improved transport models. At all scales, GOSAT generally obtains the best results of the three observing systems.

## 1   Introduction

As the second most important anthropogenic greenhouse gas after carbon dioxide in terms of radiative forcing, methane ($CH_4$) is an important climate driver. Monitoring atmospheric $CH_4$ concentrations and their driving emissions are therefore primary research objectives for Earth observation science. These two objectives are combined in atmospheric inversion systems. Such systems infer the space-time variations of the global or regional emissions from the assimilation of observations of atmospheric mole fractions into chemistry-transport models (CTMs) (Houweling et al., 1999; Bergamaschi et al., 2007; Bousquet

et al., 2011; Pison et al., 2013). For these systems, explaining the trends of $CH_4$ concentrations, such as their stability between 2000 and 2006 and their later increase (Kirschke et al., 2013), is a major scientific objective. Despite considerable efforts in developing observing systems at the Earth's surface, in the atmosphere and from space, the inverted fluxes are associated with large uncertainties. This still allows diverging interpretations of the trends, depending on which CTM is used or on how the





inversion set-up is defined (Bousquet et al., 2006, 2011; Rigby et al., 2008; Dlugokencky et al., 2009; Bergamaschi et al., 2013). In principle, the Bayesian framework should reconcile all well-tuned inversion systems because it characterizes the uncertainty of each inversion product at all space-time scales, thereby weighting each scenario suggested by the inversion approach. In practice, posterior uncertainties are often difficult to compute and are also affected by mis-specified prior or

5 observation uncertainties (Berchet et al., 2015). In a previous study, Cressot et al. (2014) applied objective tuning methods imported from Numerical Weather Prediction (Desroziers et al., 2005) within a robust Monte-Carlo approach to optimize the input error covariance matrices of a global $CH_4$ inversion system. Here, we use their results as a starting point to characterize the uncertainty of the year-to-year variations of the inverted fluxes at various temporal and spatial scales, in order to document which anomaly signals from the inversions are reliable and which are not. To do so, three different global $CH_4$ observation

systems are considered: surface sites from various global networks (flasks and continuous), the space-borne Infrared Atmospheric Sounding Interferometer (IASI) that provides a mid-to-upper-tropospheric column and the Thermal And Near infrared Sensor for carbon Observation - Fourier Transform Spectrometer (TANSO-FTS), that observes the total column from space. Using the flux anomalies of the surface inversion as the signal, signal-to-noise ratios for different temporal and spatial scales are computed, the noise being the uncertainty of the year-to-year changes of the inverted fluxes for each observing system.

Signal-to-noise ratios are then considered as a statistical criterion to evaluate the ability of an observing system to retrieve the $CH_4$ flux inter-annual variability.

The paper is structured as follows. The theoretical framework and the different data sets are presented in Section 2. The signal-to-noise ratios are presented in Section 3 and further discussed in Section 4.

## 2  Method

### 2.1  Inversion Framework

Our inversion system is based on a variational formulation of Bayes' theorem, as detailed by Chevallier et al. (2005), which has been adapted to the inversion of $CH_4$ fluxes by Pison et al. (2009). It allows inverting grid-point-scale fluxes, thereby avoiding gross aggregation errors (Kaminski et al., 2001), while assimilating the large flow of satellite data at appropriate observation times and locations. It ingests observations of $CH_4$ mole fractions and prior information about the variables that

are to be optimized, with associated error covariance matrices. Bayesian error statistics of the inverted variables are computed from a Monte-Carlo ensemble of inversions which is consistent with the assigned prior and observation errors (Chevallier et al., 2007). The inversion system includes the LMDz transport model of Hourdin et al. (2006) at resolution $3.75° \times 2.5°$ (longitude x latitude) for 19 vertical levels in a nudged and offline mode, which we couple to a simplified chemistry module (SACS) to represent the interactions between $CH_4$ and the hydroxyl radical (OH), its main sink in the atmosphere, and between

methyl chloroform (MCF) and OH. When it assimilates both $CH_4$ and MCF mole fractions, as is done here, it synergistically optimizes both $CH_4$ surface sources at weekly and model grid resolution and OH at weekly resolution over 4 latitude bands (-90/-30, -30/0, 0/30, 30/90), therefore dynamically distinguishing between $CH_4$ emission and loss. The system iteratively



minimizes the Bayesian cost function (made non-quadratic by the non-linear chemistry) using the M1QN3 algorithm (Gilbert and Lemaréchal, 1989).

This system is applied here to assimilate each one of three $CH_4$ observing systems and one MCF observing system, in the configuration used by Cressot et al. (2014). The reader is referred to Cressot et al. (2014) for a detailed description of this
configuration. It is enough here to recall that the prior fluxes (fires excepted) have no inter-annual variability (IAV). Therefore, IAV is generated from atmospheric observations and atmospheric transport and chemistry.

Two types of inversions are presented in this study: a reference inversion (hereafter called REFSURF) using $CH_4$ surface measurements from December 1999 to December 2011; and three ensembles of inversions (see Section 2.3 for the use of these), one using surface measurements (called SURF hereafter), one using IASI observations (called IASI hereafter) and one using
TANSO-FTS observations (called GOSAT hereafter), each ensemble consisting of ten inversions from 10/2009 to 09/2010.

## 2.2    Data sets

In order to have continuous and homogeneous surface data throughout the extended assimilation window, we restrict the methane site list to 36 instead of 49 as used in Cressot et al. (2014). They come from the National Oceanic and Atmospheric Administration (NOAA) global cooperative air sampling network (Dlugokencky et al., 1994, 2009), the Commonwealth Scien-
tific and Industrial Research Organisation (CSIRO) (Francey et al., 1999) and the National Institute of Water and Atmospheric Research (NIWA) (Lowe et al., 1991). We also use the station Alert (ALT) from Environment Canada (EC) (Worthy et al., 2009). MCF measurements are provided by 11 NOAA surface sites (Montzka et al., 2011). The surface sites used in our inversions are presented in Figure 1.

We use observations of the mid-to-upper tropospheric $CH_4$ column made by IASI, a thermal interferometer on-board the
Meteorological Operational (MetOp) satellites. This quantity is retrieved based on a non-linear inference scheme (Crevoisier et al., 2009) within 30 degrees of the Equator over both land and ocean at about 09:30 a.m./p.m. local time, with an accuracy of 1.2% ($\approx$20 ppb).

Last, we use observations of the $CH_4$ atmospheric total column over land from TANSO-FTS, a near-infrared spectrometer on-board the Greenhouse gases Observing SATellite (GOSAT). Total columns are retrieved by optimal estimation using the
algorithm of Parker et al. (2011) and with a precision of 0.6% ($\approx$10 ppb).

The averaging kernel or weighting function and the prior profile (when available) of each IASI or TANSO-FTS retrieval are directly accounted for in the inversion system following Connor et al. (2008).

## 2.3    Error statistics

The input error statistics for the prior and the observations are tuned using objective diagnostics as described by Cressot et al.
(2014). This means that they exhibit some objectivity that is seen to translate into realistic Bayesian posterior error statistics, which in particular make all present inversions statistically consistent at the annual and global or regional scales (Cressot et al., 2014). In order to keep the computational burden to a reasonable level, we compute the posterior error statistics from a Monte-Carlo inversion ensemble of 10 times one year (10/2009 to 09/2010). Therefore, posterior error statistics of inter-annual





emissions are computed from the ones of annual emissions by applying an inflation factor of $\sqrt{2}$. This means that we consider that the errors are uncorrelated from one year to the next. This is a conservative hypothesis since in reality some of the transport and retrieval errors are recurrent, thereby inducing positive correlations and reducing the inflation factor.

The variability of $CH_4$ concentrations depends on the oxidizing capacity of the atmosphere, which is largely controlled
by OH concentrations. Since OH concentrations are constrained through MCF data in our multi-species inversion system (Section 2.1), the uncertainty on OH ($\approx$5% after optimization) is accounted for in the uncertainty of the inverted $CH_4$ emissions and of their inter-annual variations.

## 2.4  Evaluation criterion

$CH_4$ regional flux anomalies are defined here as the deviation from the 2004-2005 mean of the $CH_4$ inverted fluxes. 2004-2005
has been chosen as a reference because it corresponds to a period of minimum atmospheric growth rate (Dlugokencky et al., 2011). The regional scale is based on the regions shown in Figure 2 and large latitudinal bands are defined as BorN for latitudes higher than 60 degrees North, MidN between 30 and 60 degrees North, TropN between 0 and 30 degrees North, TropS between 0 and 30 degrees South, MidS between 30 and 60 degrees South and BorS higher than 60 degrees South. We study various timescales from the week to 3 years.

Our criterion consists in evaluating the ability of the observing systems to detect $CH_4$ anomalies of a given amplitude, defined by the reference inversion. The inversion of surface measurements is chosen to provide the signal as the data cover a long time window (2000-2011) as compared to the two other observing systems. This longer window makes it possible to sample the $CH_4$ IAV more robustly than a 2-3 year inversion. We compare the $CH_4$ anomalies derived from REFSURF to the error variances computed for each observing system (from SURF, IASI and GOSAT). The Bayesian posterior error variances
associated with the IAV of $CH_4$ fluxes are computed from the Monte-Carlo ensemble as described in Section 2.3 and constitute the noise associated to each observing system. To evaluate the space-time scales at which the anomalies are larger than the detection limit of each observing system, the signal-to-noise ratios are computed at the same spatial scales for weekly to 3-yearly time scales. This statistical criterion estimates for which time scales and regions the $CH_4$ anomalies are reliable for each observing system. In the following, the presentation of the results is done for three timescales (seasonal, yearly, and 3-yearly
trends) before assessing their sensitivity to temporal and spatial aggregations.

## 3  Results: signal-to-noise ratios

### 3.1  Seasonal-scale detection

The signal-to-noise ratios are computed over three-month periods (JFM, AMJ, JAS and OND, hereafter referred to as "seasons" for simplicity) from 2000 to 2011 i.e. 48 occurrences (12 years of 4 seasons). The three observing systems are able to detect
almost all the anomalies at the global scale (Table 1). As expected, the fraction of detected anomalies decrease with the spatial scale. At the global scale, 93 to 97% of the flux anomalies are detected depending on the observing system (Table 1). At



semi-hemispheric scales (excluding MidS and BorS areas), this range is of 10-91% (median = 52%), GOSAT having the best range (25-91%) compared to IASI (22-79%) and SURF (10-81%). The lack of detection in MidS and BorS is not significant considering the small methane fluxes involved. At the regional scale, the detection range is 0-97% (median = 10%), with large contrasts. Again the range is more favorable for GOSAT (0-97%, median = 20%) than for SURF (0-87%, median = 10%) and IASI (0-52%, median = 6%). Only anomalies in Central America are not detected by any of the three observing systems. GOSAT is the only one of the three observing systems to detect any anomalies in the USA, temperate South America [SouthSAm] and temperate Africa [SouthernAfr].

At the seasonal time-scale, large signals are due to various causes, depending on the emitting area. At high Northern latitudes, a large seasonal cycle is expected for wetland emissions, with mostly no emissions during winter and maximum emissions during summer: this leads to four seasons very different from their average and therefore to large anomalies. This is illustrated on the detection in NorthAmBor (Table 1): GOSAT is able to detect almost all the anomalies, half of which are positive (Table 1/Figure 3, due to maximum emissions in spring and summer) and half negative (Table 1/Figure 3, due to almost null emissions when the surface is snow-covered). Due to a larger noise ($\approx$1.4 Tg vs $\approx$1 Tg for GOSAT, Figure 4 [a]), SURF misses some springs (Figure 3); and IASI, with the largest noise ($\approx$1.9 Tg, Figure 4 [a]), mostly detects winter and summer (Figure 3). In the larger BorN area, only winter and summer are detected (Figure 3).

In the Tropics, some areas also have large seasonal variations, mainly due to biomass burning or rice-paddies. In AfrEquat, the AMJ positive signals generated are almost all detected by GOSAT (Figure 4 [a]). Note that SURF performs poorly in this area (Table 1), due to the lack of stations which leads to large noise ($\approx$3.3 Tg, Figure 4 [a]). In India and China, the rice-paddy practices lead to a seasonal cycle of methane emissions with a maximum in JAS and a minimum in JFM (Matthews et al., 1991). The three systems detect anomalies in JFM and JAS (Figure 3) with consistent signs (half positive, half negative anomalies for GOSAT and IASI, positive anomalies preferentially detected by SURF (Table 1).

## 3.2 Yearly-scale detection

The signal-to-noise ratios are computed over the years from 2000 to 2011 i.e. 12 occurrences. At the yearly scale, detection rates are smaller than at the seasonal scale, at all spatial scales. Note that most anomalies are positive since the reference for computing the signal is 2004-2005 i.e. the period of global minimum over 2000-2011. At the global scale, detection rates range from 58% to 83% (Table 2). The Boreal zone [BorN] is only poorly detected (8%) whereas the Tropics [TropN and TropS] remain the best detected zone (16-58%). At the regional scale, the detection rates range between 0 and 58% with a median of 0%: the only regions above 25% of detection are Africa [NorthAfrWest, NorthAfrEast, AfrEquat], Middle East for GOSAT and Eastern Siberia [FarEastSib]. No detection is obtained in key regions for methane emissions such as Amazonia, India, China (except SURF at 16%) and North America [NorthAmBor, USA].

The differences between the three observing systems are larger at the yearly scale than at the seasonal scale: GOSAT and IASI detect more than 75% of the 12 possible global occurrences versus 58% for SURF (Table 2). At the regional scale, GOSAT detects more anomalies than the two other systems, IASI and SURF being comparable in their detection rates. Indeed, GOSAT noises are smaller than the two other systems (<1.5 Tg in MiddleEast for GOSAT against >4.5 Tg for SURF and



IASI; <1.8 Tg in NorthAfrEast for GOSAT against >2.9 Tg for SURF and IASI). This is partly due to the large number of data available in these two regions (Table 4): with NorthAfrWest, CentralAsia and AustrNZ, they have the largest number of GOSAT data, mainly because they are among the driest areas i.e. with the lowest cloud cover. In agreement with the intuition of Bergamaschi et al. (2013) that performing gross averages makes it possible to extract a signal from the inversion, the detection

is enhanced in the latitudinal bands e.g. detection rates >50% in MidN and TropN for GOSAT, TropN for SURF. But, at the regional scale, it remains difficult to robustly extract yearly flux anomalies. Therefore, we now focus our analysis on longer time scales, with a longer time aggregation of three years, to get hints at the trends in methane emissions.

### 3.3   Trend detection over 2000-2011

Aggregating through time while still retaining a small enough resolution to discuss tendencies over 2000-2011, we define four

time-windows of three years each: 2000-2002, 2003-2005, 2006-2008 and 2009-2011. The reference period for the definition of the anomalies of each of these four periods is still 2004-2005 (Section 2.4).

At the global scale, the emissions have slowly decreased from 2000 to 2005, with a global minimum in 2004-2005, then increased at a larger rate after 2006 (Kirschke et al., 2013). The three observing systems are able to detect the large positive anomalies after 2006 and consistently detect nothing or small positive anomalies before (Table 3). The three observing systems

are able to detect the same time-evolution of the signal in TropN. Only GOSAT and SURF detect MidN anomalies; the lower detection by IASI at these latitudes is expected since the data used here are only within +/-30 degrees of the Equator (Table 4: no IASI data in MidN). The signal in BorN is never detected. This is consistent with the recent increase of methane global emissions coming mostly from the Tropics and to a lesser extent from the northern mid-latitudes, as suggested by Bergamaschi et al. (2013) and Nisbet et al. (2014).

Being able to detect anomalies at a smaller spatial scale could help attributing the changes in methane emissions to particular processes. Unfortunately, even when aggregating 3 years together (instead of one as in Section 3.2), it is still difficult to detect regional anomalies. On top of the regions already detected at the yearly time scale, a positive change in Chinese emissions is detected with the three-year aggregation, but only by IASI and SURF. The lack of detection by GOSAT stems from the small number of GOSAT data compared to IASI over India and China (Table 4), which is due to cloud cover and aerosol column

content. On the contrary, GOSAT alone suggests detectable negative anomalies in NorthAmBor in 2000-2002 and 2009-2011. This is consistent with the lack of surface sites in this area (e.g. the Canadian stations from Environment Canada were not used here) and the lack of data by IASI North of 30 degrees (Table 4).

At high northern latitudes, positive anomalies in FarEastSib are detected by all three systems in 2000-2005 and again in 2009-2011 by GOSAT and SURF, even though the emissions in this area are small (1 Tg in 2004-2005, Table 3). This is due

to the very small noises, mainly due to the small prior errors, which are built proportional to the fluxes. Moreover, for SURF, 3 stations are available downwind of this region.

In TropN, among the regions with a good detection rate are NorthAfrWest and NorthAfrEast plus part of AfrEquat, the remainder of this region being in TropS. In these regions, all three observing systems detect anomalies, even though GOSAT has the largest signal-to-noise ratios. Note that SURF seems to be able to make use of the stations located mostly on the coasts





(only ASK is actually in the land-mass). GOSAT is also able to detect large negative (2000-2003) and positive (2006-2011) anomalies in the MiddleEast; SURF is under the detection threshold because the available station in the region, WIS, is upwind of the area and no other station is available close enough downwind; the anomalies are not detected by IASI either because IASI's weight-function peaks in the mid-troposphere. In a region dominated by subsidence, like the MiddleEast, the altitude concentrations seen by IASI are not directly connected to the surface. The detection of surface variations in the fluxes is therefore poor, contrary to regions dominated by convection like Indonesia, where IASI has the best detection rates. In China, the three systems agree on the detectable negative anomalies in 2000-2002 and do not detect any signal in 2003-2005. After 2006, SURF detects the positive anomalies, because its noise is the smallest ($\approx$12 Tg) with about 3 stations providing relatively direct constraints in the region. The two satellites, for which noises are 15 to 40% larger ($\approx$14 and $\approx$17 Tg), do not detect this signal.

In Indonesia, IASI and GOSAT agree on detectable positive anomalies in 2000-2002 and 2006-2008 and nothing detectable for the other two periods. Indeed, no large El Niño occurred during the first decade of the 21st century with the associated large fires such as those experienced late in 2015 for instance (National Weather Service - Climate Prediction Center, 2016).

Among the key-areas for methane emissions, signals in Amazonia (dominated by tropical wetlands) and in BorN, particularly in SiberianLowlands (dominated by boreal wetlands in summer), remain undetectable by the three systems. In SiberianLowlands, the noises of the three systems are small (between 3 and 6 Tg [not shown]); in Amazonia, the noises of the satellites are relatively small ($\approx$8 and $\approx$6 Tg resp. for GOSAT and IASI), whereas the noise of SURF, for which no stations are available closer than ASC in the Atlantic, is $\approx$19 Tg (Figure 7, 3Y case). Nevertheless, all these anomalies remain smaller than the smaller noise, and are therefore not detectable. This is because the signal variability remain small after inversion (less than 20% of the average mass over 2004-2005). As there is no IAV in the prior emissions (except biomass burning), the lack of constraints from the atmosphere leads some fluxes to stick to the low-IAV prior, leading to small anomalies.

### 3.4 Detection at other timescales

As shown previously, the temporal scale at which the signal and noise are computed has an impact on the detection. Section 3.1 deals with the 3-monthly time-scale over a 12-year time-window; Section 3.3 deals with the 3-yearly time-scale in 3-year time-windows. The impact of temporal aggregation on the noise and the signal in these time-windows is displayed in Figure 5, Figure 6 and Figure 7 for three areas: Global, hemispheric with the example of BorN, and regional with the example of Amazonia. At all spatial scales, the noises and signals are smaller when the time-scale is smaller (from 3-yearly to weekly). As expected for emissions with "seasonal" cycles, the seasonal scale (4- or 3-monthly) is particularly detected (Figure 5, Figure 6) in our relatively large areas.

The minimum time-resolution of one week could be relevant in regions where the signal is mostly from wetland emissions and/or biomass burning; it would be useful to be able to detect the beginning of the emitting season for wetlands and the short-lived fires. In NorthAmBor, where both these sources are found, about 55% of weekly anomalies are detected by GOSAT and SURF. In all other regions, the detection rates at this time-scale are small ($<\approx$25%, not shown).





In key-region Amazonia (Figure 7), no signal is detected at the 3-yearly time-scale nor at the weekly time-scale by any of the three systems; only GOSAT detects about 15% of the anomalies at the yearly time-scale. Actually, the time-scale at which the best detection rates are found depends on the region and varies from the largest possible (12-year scale) to the 2-month scale. In most of Africa [NorhtAfrWest, NorthAfrEast, AfrEquat], the signal at the 12-year scale is detected by all three systems (it

is detected by GOSAT only in SouthernAfr). In India and China, the best detection rates are obtained at the 2- or 3-monthly time-scale for GOSAT (77 and 43% respectively) but at the 4-monthly time scale for IASI (66 and 30% respectively) and SURF (61 and 44% respectively). At high latitudes [BorN], the best detection rates are found at the 2-monthly time-scale (between 70% for IASI and 86% for GOSAT).

    In order to further understand the various levels of detection described above, we investigate the sensitivity of our results to

two main parameters of our set-up: spatial aggregation and signal used.

## 4   Sensitivity analysis

### 4.1   Impact of spatial aggregation on trend detection

Our inversion systems solves methane fluxes at the model resolution (3.75°x2.5°) worldwide. Although spatial and temporal correlations are prescribed (see Section 2.3), flux anomalies of different signs may still be obtained. These anomalies may be

either the realistic result of the constraints or due to the optimization taking an easy path when too few constraints are available. The definition of larger areas may lead to summing up anomalies of opposite signs and hide (realistic or not) spatial variations. We then try to investigate the impact of the spatial aggregation of model pixels in the case of one illustrative region, Amazonia, which is a key-area for methane emissions and remains poorly detected by all the studied observing systems at all time-scales (see Section 3.4). In the region as defined on our model grid, the signal at the pixel scale is indeed patchy (Figure 8). Dipoles

of negative/positive signal are summed up when aggregating at the region's scale. The impact of the progressive aggregation of rings of pixels from the center of Amazonia is displayed in Figure 9: the 3-yearly signal could be detected by all systems for the four 3-year periods up to the 3rd ring i.e. for a region covering 25 pixels instead of 66. It would then be possible to define the regions based on the spatial aggregation that allows the best detection rates for the chosen observing system. This would nevertheless lead to the issue of the user needs e.g. whether the regions are actually relevant for country budgets.

### 4.2   Impact of the signal on seasonal and yearly detection

Since the signal is obtained from one inversion only i.e. depends on numerous assumptions (error statistics, set of assimilated data, etc) and has potentially large uncertainties in various areas (e.g. far from the observing stations), another signal definition has been tested. It must cover enough years of analysis to be representative of the variability of methane fluxes. We therefore chose an inversion by Bousquet et al. (2011), (called PBSURF hereafter) instead of the SURF inversion described above.

PBSURF solves methane fluxes for large regions and several processes in each region, using observations from a set of surface stations different from SURF. The large-region-scale inversion means that the spatial variability of the prior is kept within each



region and is only scaled (contrary to SURF, which is performed at the pixel scale i.e. is able to vary only a few pixels to match the data). This difference in the methods may lead to very different spatial variability in each of the regions of interest (Figure 4), a larger variability ensuring a better detection rate with our criterion.

We first focus on the seasonal (3-monthly) scale, which is the time-scale at which the detection is the most favorable for
SURF (Section 3.4) while being relevant for methane emissions at the regional scale defined here. The issue here is not whether the two inversions agree on the retrieved fluxes but whether the detection rates differ. Europe illustrates how the detection rates of two signals can differ: for all three observing systems, PBSURF signal is more than twice as often detected as SURF and the signs of the detected anomalies are opposite (positive for SURF, negative with PBSURF, Table 1 and Table 5: less positive anomalies are detected for a larger total number of detected anomalies).

The signal by PBSURF contains more negative anomalies than SURF at the global scale and in BorN and MidN (for GOSAT and SURF). This is due to the fact that the two years of global minimum in SURFPB are not 2004 and 2005 but 2004 and 2006, so that using 2004-2005 as the reference period does not lead to mainly positive anomalies. For the three observing systems, detection is better with PBSURF signal in the Southern hemisphere (TropS, MidS). In the Northern hemisphere, at the regional scale, the detection rate is shifted in longitude. NorthAmBor seasons are about half as often detected whereas
up to 5 times more occurrences are detected in SiberianLowlands, SiberianHighlands and FarEastSib. In SiberianLowlands and FarEastSib, the larger number is due to negative signals for GOSAT and SURF. The same pattern is seen in the mid-latitudes where MiddleEast, India and China, which are almost never detected with PBSURF signal, versus NorthAfrWest and NorthAfrEast, in which mainly positive anomalies are detected (IASI and SURF) or both positive and negative anomalies (GOSAT). The regional scale in the Southern hemisphere confirms the better detection with PBSURF signal (Amazonia,
SouthSAm, SouthernAfr, Indonesia, AustrNZ). In Amazonia, the (mainly positive) signals are detectable by GOSAT and IASI, but China (resp. India) is not anymore (resp. poorly) detectable using PBSURF.

At the yearly scale (Table 6), the detection rates are shifted to the North in the Northern hemisphere and to the South in the Southern hemisphere (from TropN and MidN to BorN and TropS). Detection rates higher than 50% are found in Amazonia for GOSAT and IASI; in Europe for GOSAT and SURF; in Indonesia for GOSAT and IASI.

One important outcome of this sensitivity test to the signal is that regional or hemispheric flux anomalies are detected at all latitudes at most time-scales but the localization of the detected signal varies depending on the inversion characteristics (including the observations used). This is of course one important limitation in attributing the observed atmospheric changes to particular regions and to the underlying emission processes.

The impact of the signal on the detection of anomalies has also been tested by using a variational inversion at the pixel scale
assimilating both surface and IASI data. With this signal, the detection rates are higher in the Tropics (particularly in India and China) and in the Southern hemisphere at mid-latitudes [not shown]. This suggests that the joint assimilation of surface and satellite data may lead to a better localization of the anomalies of the surface methane fluxes. Nevertheless, this requires that the consistency between the two types of data (surface and remote-sensed) be improved (Locatelli et al., 2015; Monteil et al., 2013).



## 5 Conclusions

The aim of this study was to investigate which spatial and temporal scales current atmospheric inversions may detect in terms of methane surface flux anomalies. To do so, we have proposed a signal-to-noise ratio analysis, the signal being the methane fluxes inferred from a reference surface-based inversion from 2000 to 2011 and the noise being computed from three inversion systems using surface or satellite data (GOSAT and IASI). At the global and semi-hemispheric scales, all observing systems detect flux anomalies at all time-scales from seasonal (3-month average) to long-term trend (3-year average). At all scales, GOSAT generally shows the best results among the different systems.

At the regional scale, the results are more variable. The seasonal changes are all detected with fair to good rates by at least one network (GOSAT), and more than 50% of the regions are detected by the three networks. The year-to-year changes and longer term trends (three year averages) are detected for up to 50% of the regions (by GOSAT) with detection rates mostly lower than 50%. Anomalies in African regions (all), Middle East (GOSAT), Eastern Siberia and Europe (all) are detected with variable rates. In some key regions for the methane cycle, anomalies are hardly detected, both in the case of dominant anthropogenic emissions (North America) or natural emissions (Amazonia, Siberian lowlands). A sensitivity test to the spatial scale through aggregation shows that dipole effects in the retrieved flux anomalies prevent anomalies in Amazonia (as defined in this study) to be detected. Flux anomalies in India and China, two areas with large and mixed (natural and anthropogenic) methane emissions, are generally poorly detected. Only a long-term trend over China is detected, with larger emissions after 2006 for IASI and the surface network but not for GOSAT, which has a lower number of observations over these regions because of cloud cover and aerosol layers. North-American emission changes are not detected, with the exception of the long term trend of boreal North America (negative after 2006 for GOSAT). Overall, the detection at a yearly scale is generally poor to fair for the two signals tested, both obtained with surface constraints. This suggests that the ability of the inversions to retrieve significant inter-annual variations in the methane fluxes is not evident and should be evaluated against uncertainties, which are not always computed and/or provided with the inversion products.

The use of another signal (a different surface-based inversion) does not change the main conclusion that anomalies at the regional scale are only fairly-well detected but shows that the regions which are not seen may be different: some yearly changes in Amazonia and India can be detected but tropical Africa is much less detected with the second signal. Therefore, the precise identification of flux anomalies in the Tropics appears not to be robust with regards to changes in the inversion used for the signal. This is of course an issue when attributing the increase observed in atmospheric methane since 2006 to a particular region, as already noticed by Locatelli et al. (2015).

To increase the detection rates, the number of constraints (i.e. of assimilated data, either from satellite or from surface sites) should be increased, as shown by the regional differences between the two surface-based inversions (e.g. Africa versus Tropical regions and China) and between the satellite based inversions (more IASI observations over China and India than GOSAT ones). To increase the robustness of the attribution of flux anomalies to a particular region, transport models should be improved together with the consistency of error statistics prescribed for flux and observations (Berchet et al., 2015).Defining smaller regions, as tested here in Amazonia, may also improve the detection of anomalies in small key-areas with intense





methane emissions. The joint assimilation of surface and satellite observations could be a solution to better constrain the surface methane fluxes, if the consistency between surface and remote sensed data can be improved (Locatelli et al., 2015; Monteil et al., 2013). Cloud cover and aerosol layers may limit the observability of key regions such as China and India. Solar based satellite instruments also provide limited data at high latitudes. The future space mission MERLIN, based on a

5 differential LIDAR measurement with a very small spot on the ground, is less sensitive to cloud cover and does not need light to provide data (Kiemle et al., 2014). In this context, MERLIN seems a promising mission to improve some of the limitations raised in this paper.

*Acknowledgements.* The authors are very grateful to the many people involved in the surface and satellite measurement and in the archiving of these data. The authors particularly thank E.J. Dlugokencky (NOAA), S.A. Montzka (NOAA), C. Crevoisier (LMD), H. Boesch (University

of Leicester), R. Parker (University of Leicester), P.B. Krummel (CSIRO), L.P. Steele (CSIRO), R.L Langenfelds (CSIRO), S. Nichol (NIWA) and D. Worthy (EC). We aknowledge the contributors to the World Data Center for Greenhouse Gases for providing their data of methane and methyl-chloroform atmospheric mole fractions. The first author is funded by CNES and CEA. P. J. Rayner is in receipt of an Australian Professorial Fellowship (DP1096309). This work was performed using HPC resources from CCRT under the allocation 2014-t2014012201 made by GENCI (Grand Equipement National de Calcul Intensif) and a DSM allocation. We also thank the computing support team of the

LSCE led by F. Marabelle.





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



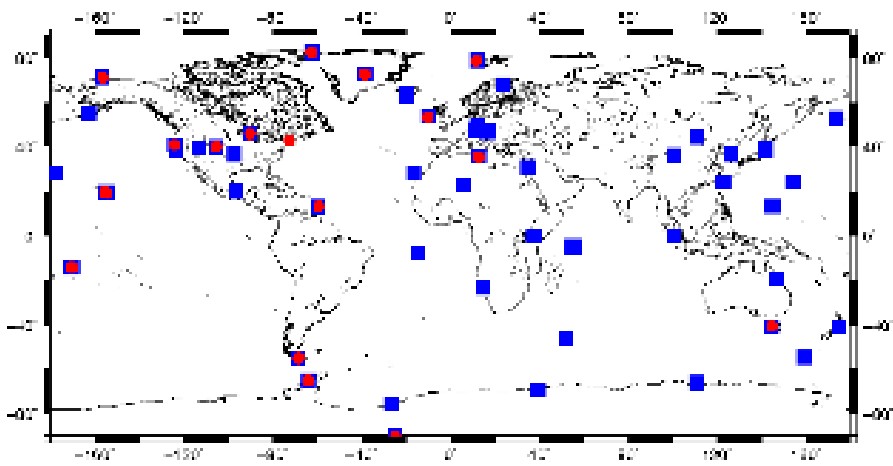

**Figure 1.** *Surface sites from the NOAA, CSIRO, NIWA and EC networks used in this study with red circles for surface sites observing MCF dry air mole fractions and blue squares for surface sites observing CH$_4$ dry air mole fractions.*

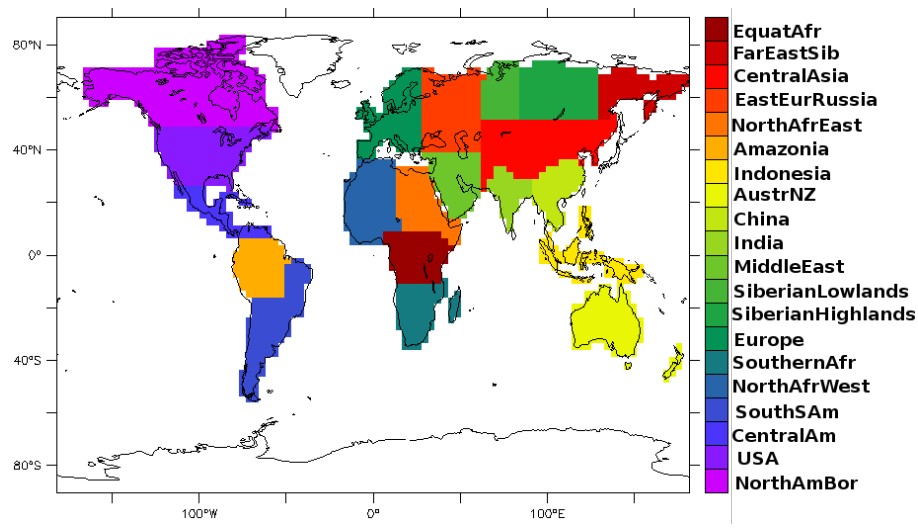

**Figure 2.** *Regions on the model grid, adapted to key-area for methane fluxes.*



**Figure 3.** *Number of detected seasons over the 12 possible for winter (JFM, blue), spring (AMJ, green), summer (JAS, red) and fall (OND, orange) in the various regions.*





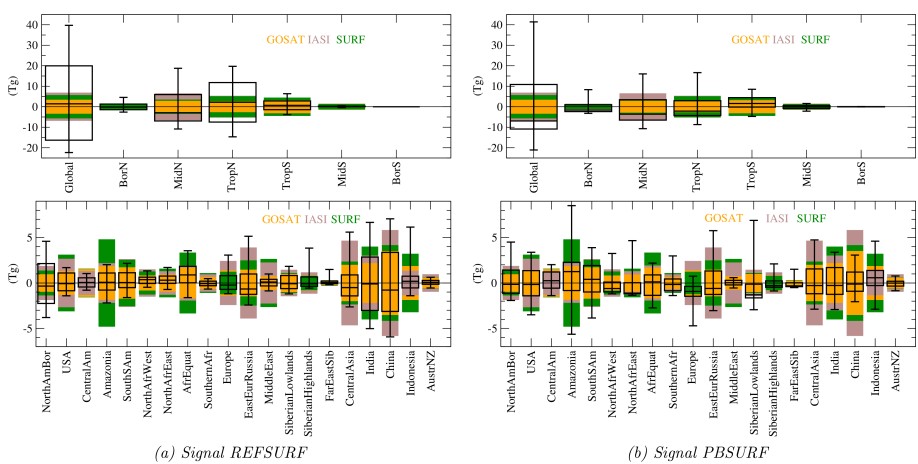

**Figure 4.** *Noise by the three observing systems (bars) and box plots (median, 25 and 75%) for the signal in various areas (latitudinal bands and regions). Detection is achieved when the signal is larger than the noise i.e. for all the occurrences in each box plot which lay outside the matching colored bar.*

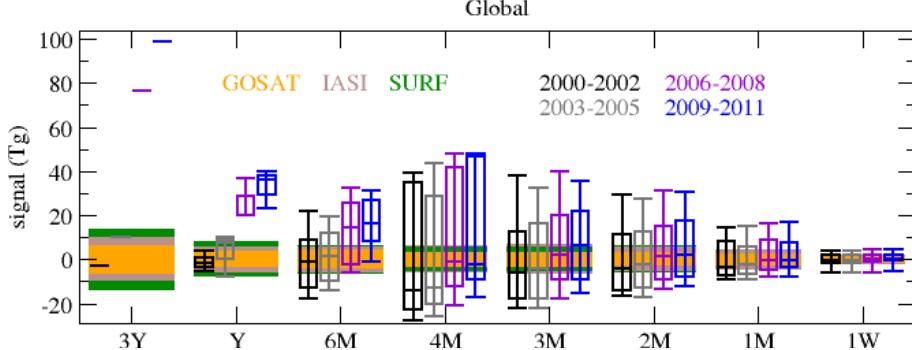

**Figure 5.** *Impact of temporal agregation on noise and signal over 3-year time-windows. Link to Table 3: the Global lines of the Table corresponds to the 3Y bars here.*




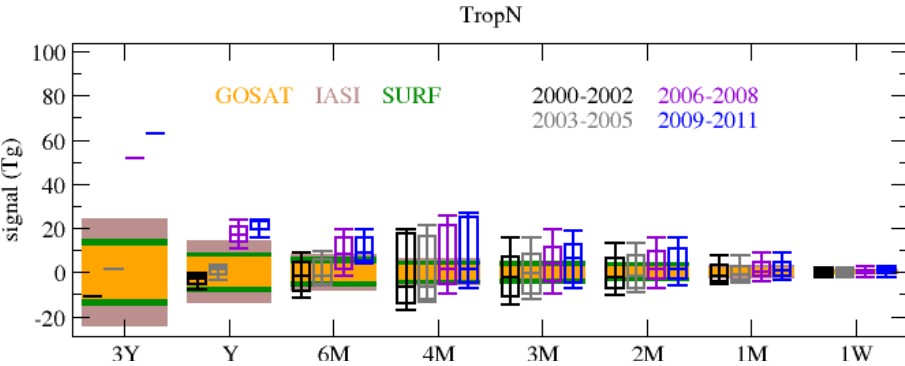

**Figure 6.** *Impact of temporal agregation on noise and signal over 3-year time-windows. Link to Table 3: the TropN lines of the Table corresponds to the 3Y bars here.*

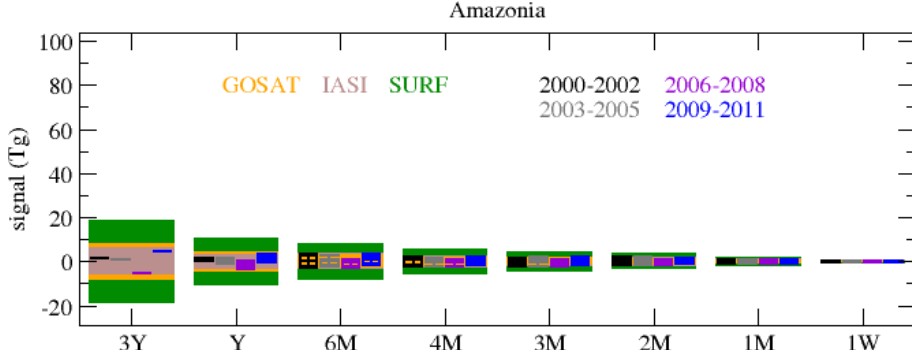

**Figure 7.** *Impact of temporal agregation on noise and signal over 3-year time-windows. Link to Table 3: the Amazonia lines of the Table corresponds to the 3Y bars here.*



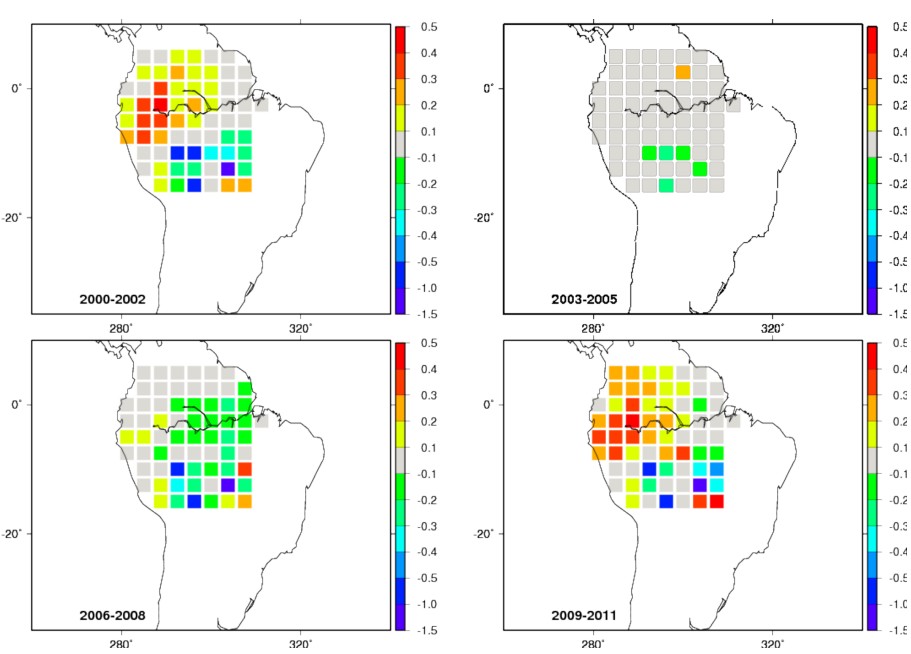

**Figure 8.** *Signal (Tg) for the four 3-year time-windows at the pixel scale.*

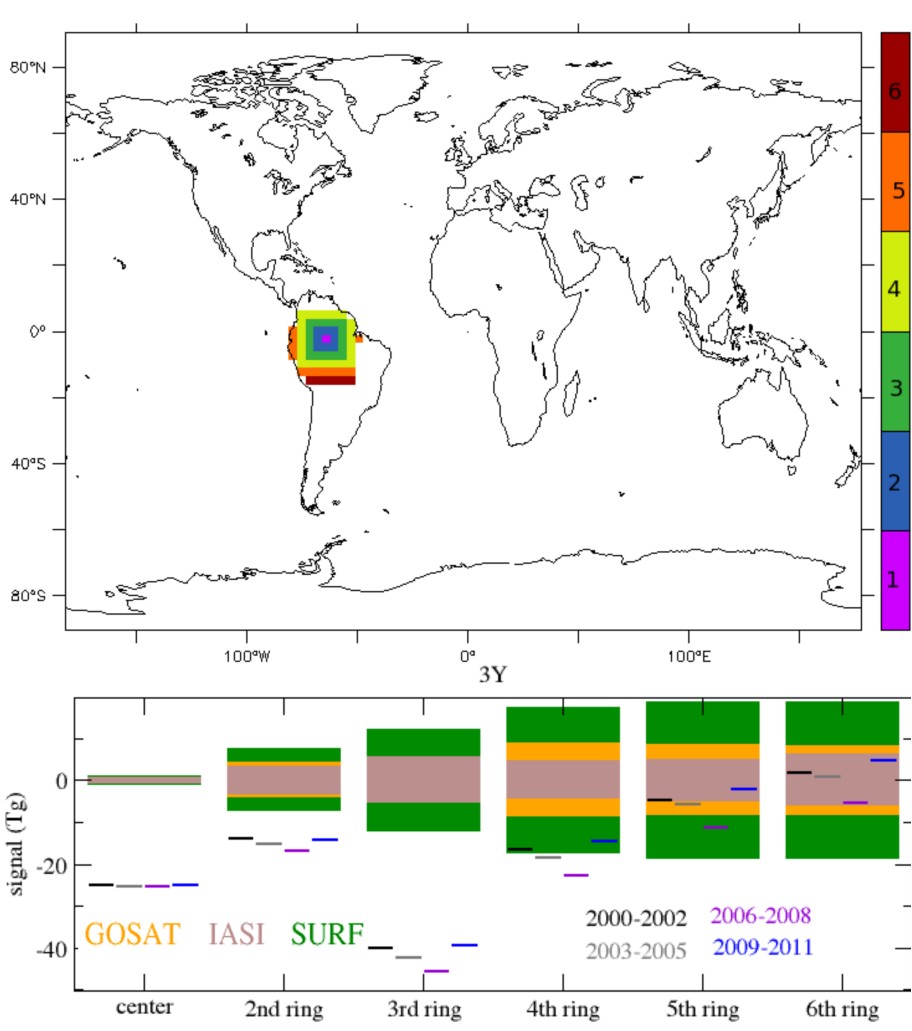

**Figure 9.** *Impact of spatial aggregation in Amazonia: from a unique pixel to larger rings around it.*





Table 1: Detection of the signal consisting in the anomalies at the "seasonal" time-scale i.e. quarters of the year (JFM, AMJ, JAS, OND). The signal is the difference between each quarter in the 2000-2011 period (i.e. 48 occurrences) and the 2004-2005 average from REFSURF. The noise is computed at the quarter time-scale from each of the three observation systems, GOSAT, IASI and SURF. See Section 2.4 and Section 2.3 for details. In each cell of the Table, we show X%(YY/ZZ) where X% is the percentage of quarterly anomalies detected (among 48 possible), YY is the number of positive anomalies detected among the ZZ detected anomalies. Column "Ave. mass" indicates the average emitted mass of $CH_4$ over 2004-2005 in the area.

| Region | Ave. mass (Tg) | GOSAT | IASI | SURF |
|---|---|---|---|---|
| Global | 517 | 97%(24/47) | 93%(22/45) | 93%(22/45) |
| BorN | 18 | 50%(12/24) | 50%(12/24) | 52%(13/25) |
| MidN | 177 | 87%(18/42) | 54%(12/26) | 81%(16/39) |
| TropN | 194 | 91%(24/44) | 79%(20/38) | 81%(21/39) |
| TropS | 115 | 25%(10/12) | 22%(10/11) | 10%(05/05) |
| MidS | 12 | ∅ | ∅ | ∅ |
| BorS | 1 | ∅ | ∅ | ∅ |
| NorthAmBor | 20 | 97%(23/47) | 52%(13/25) | 87%(18/42) |
| USA | 37 | 20%(09/10) | ∅ | ∅ |
| CentralAm | 17 | ∅ | ∅ | ∅ |
| Amazonia | 38 | 08%(01/04) | 02%(00/01) | ∅ |
| SouthSAm | 30 | 06%(03/03) | ∅ | ∅ |
| NorthAfrWest | 13 | 16%(08/08) | ∅ | 04%(02/02) |
| NorthAfrEast | 11 | 20%(10/10) | 06%(03/03) | 02%(01/01) |
| AfrEquat | 32 | 35%(17/17) | 20%(10/10) | 04%(02/02) |
| SouthernAfr | 10 | 04%(00/02) | ∅ | ∅ |
| Europe | 33 | 14%(07/07) | 04%(02/02) | 14%(07/07) |
| EastEurRussia | 30 | 45%(12/22) | 08%(04/04) | 22%(11/11) |
| MiddleEast | 16 | 14%(04/07) | ∅ | ∅ |





Table 1: (continued) Detection of the signal consisting in the anomalies at the "seasonal" time-scale.

| Region | Ave. mass (Tg) | GOSAT | IASI | SURF |
|---|---|---|---|---|
| SiberianLowlands | 8 | 47%(12/23) | 12%(06/06) | 35%(12/17) |
| SiberianHighlands | 5 | 22%(11/11) | 06%(03/03) | 22%(11/11) |
| FarEastSib | 1 | 16%(08/08) | 16%(08/08) | 16%(08/08) |
| CentralAsia | 28 | 33%(09/16) | 02%(01/01) | 18%(06/09) |
| India | 50 | 62%(13/30) | 50%(12/24) | 35%(12/17) |
| China | 64 | 43%(11/21) | 10%(03/05) | 31%(09/15) |
| Indonesia | 36 | 06%(03/03) | 16%(07/08) | 06%(03/03) |
| AustrNZ | 6 | 02%(01/01) | $\varnothing$ | 02%(01/01) |





Table 2: Detection of the signal consisting in the anomalies at the yearly time-scale. The signal is the difference between each year in the 2000-2011 period (i.e. 12 occurrences) and the 2004-2005 average from REFSURF. The noise is computed at the yearly time-scale from each of the three observation systems, GOSAT, IASI and SURF. See Section 2.4 and Section 2.3 for details. In each cell of the Table, we show X%(YY/ZZ) where X% is the percentage of yearly anomalies detected (among 12 possible), YY is the number of positive anomalies detected among the ZZ detected anomalies. Column "Ave. mass" indicates the average emitted mass of $CH_4$ over 2004-2005 in the area.

| Region | Ave. mass (Tg) | Gosat | Iasi | Surf |
|---|---|---|---|---|
| Global | 517 | 83%(08/10) | 75%(08/09) | 58%(07/07) |
| BorN | 18 | 08%(01/01) | ∅ | 08%(01/01) |
| MidN | 177 | 66%(07/08) | ∅ | ∅ |
| TropN | 194 | 58%(06/07) | 41%(05/05) | 50%(06/06) |
| TropS | 115 | 25%(03/03) | 33%(04/04) | 16%(02/02) |
| MidS | 12 | ∅ | ∅ | ∅ |
| BorS | 1 | ∅ | ∅ | ∅ |
| NorthAmBor | 20 | ∅ | ∅ | ∅ |
| USA | 37 | ∅ | ∅ | ∅ |
| CentralAm | 17 | ∅ | ∅ | ∅ |
| Amazonia | 38 | ∅ | ∅ | ∅ |
| SouthSAm | 30 | 08%(01/01) | ∅ | ∅ |
| NorthAfrWest | 13 | 41%(05/05) | 33%(04/04) | 41%(05/05) |
| NorthAfrEast | 11 | 50%(06/06) | 25%(03/03) | 08%(01/01) |
| AfrEquat | 32 | 41%(05/05) | 33%(04/04) | 33%(04/04) |
| SouthernAfr | 10 | ∅ | ∅ | ∅ |
| Europe | 33 | 16%(02/02) | 08%(01/01) | 16%(02/02) |
| EastEurRussia | 30 | ∅ | ∅ | ∅ |
| MiddleEast | 16 | 58%(04/07) | ∅ | ∅ |





Table 2: (continued) Detection of the signal consisting in the anomalies at the yearly time-scale.

| Region | Ave. mass (Tg) | Gosat | Iasi | Surf |
|---|---|---|---|---|
| SiberianLowlands | 8 | ∅ | ∅ | ∅ |
| SiberianHighlands | 5 | 08%(01/01) | 08%(01/01) | 08%(01/01) |
| FarEastSib | 1 | 25%(03/03) | 25%(03/03) | 25%(03/03) |
| CentralAsia | 28 | 08%(00/01) | ∅ | ∅ |
| India | 50 | ∅ | ∅ | ∅ |
| China | 64 | ∅ | ∅ | 16%(00/02) |
| Indonesia | 36 | 16%(02/02) | 25%(02/03) | 16%(02/02) |
| AustrNZ | 6 | 16%(02/02) | ∅ | ∅ |





Table 3: Detection of the signal consisting in the anomalies at the 3-yearly time-scale. The signal is the difference between each 3-year time-window in the 2000-2011 period (2000-2002, 2003-2005, 2006-2008, 2009-2011) and the 2004-2005 average from REFSURF. The noise is computed at the 3-yearly time-scale from each of the three observation systems, GOSAT, IASI and SURF. See Section 2.4 and Section 2.3 for details.

In each cell of the Table, we show whether a positive anomaly, a negative anomaly or no anomaly is detected and with which signal-to-noise ratio: positive anomaly detected: +++ = with stn ratio > 3, ++= stn ratio > 2 and + = stn ratio > 1; negative anomaly detected with - -= stn ratio <-2, - = stn ratio <-2,∅ = no anomaly detected.

The number below the name of the area is the average emitted mass of $CH_4$ over 2004-2005 in the area.

| Region | System | 2000-2002 | 2003-2005 | 2006-2008 | 2009-2011 |
|---|---|---|---|---|---|
| Global | Gosat | ∅ | + | +++ | +++ |
| 517 | Iasi | ∅ | + | +++ | +++ |
| | Surf | ∅ | ∅ | +++ | +++ |
| BorN | Gosat | ∅ | ∅ | ∅ | ∅ |
| 18 | Iasi | ∅ | ∅ | ∅ | ∅ |
| | Surf | ∅ | ∅ | ∅ | ∅ |
| MidN | Gosat | - | ∅ | ++ | ++ |
| 177 | Iasi | ∅ | ∅ | ∅ | ∅ |
| | Surf | ∅ | ∅ | + | + |
| TropN | Gosat | ∅ | ∅ | +++ | +++ |
| 194 | Iasi | ∅ | ∅ | ++ | ++ |
| | Surf | ∅ | ∅ | +++ | +++ |
| TropS | Gosat | + | ∅ | ∅ | + |
| 115 | Iasi | + | ∅ | ∅ | + |
| | Surf | + | ∅ | ∅ | + |
| MidS | Gosat | ∅ | ∅ | ∅ | ∅ |
| 12 | Iasi | ∅ | ∅ | ∅ | ∅ |
| | Surf | ∅ | ∅ | ∅ | ∅ |
| BorS | Gosat | ∅ | ∅ | ∅ | ∅ |
| 1 | Iasi | ∅ | ∅ | ∅ | ∅ |
| | Surf | ∅ | ∅ | ∅ | ∅ |





Table 3: (continued) Detection of the signal consisting in the anomalies at the 3-yearly time-scale.

| Region | System | 2000-2002 | 2003-2005 | 2006-2008 | 2009-2011 |
|---|---|---|---|---|---|
| NorthAmBor | Gosat | - | ∅ | ∅ | - |
| 20 | Iasi | ∅ | ∅ | ∅ | ∅ |
|  | Surf | ∅ | ∅ | ∅ | ∅ |
| USA | Gosat | ∅ | ∅ | ∅ | ∅ |
| 37 | Iasi | ∅ | ∅ | ∅ | ∅ |
|  | Surf | ∅ | ∅ | ∅ | ∅ |
| CentralAm | Gosat | ∅ | ∅ | ∅ | ∅ |
| 17 | Iasi | ∅ | ∅ | ∅ | ∅ |
|  | Surf | ∅ | ∅ | ∅ | ∅ |
| Amazonia | Gosat | ∅ | ∅ | ∅ | ∅ |
| 38 | Iasi | ∅ | ∅ | ∅ | ∅ |
|  | Surf | ∅ | ∅ | ∅ | ∅ |
| SouthSAm | Gosat | ∅ | ∅ | ∅ | + |
| 30 | Iasi | ∅ | ∅ | ∅ | ∅ |
|  | Surf | ∅ | ∅ | ∅ | + |
| NorthAfrWest | Gosat | ∅ | ∅ | + | ++ |
| 13 | Iasi | ∅ | ∅ | + | ++ |
|  | Surf | ∅ | ∅ | + | ++ |
| NorthAfrEast | Gosat | ∅ | ∅ | +++ | +++ |
| 11 | Iasi | ∅ | ∅ | + | + |
|  | Surf | ∅ | ∅ | + | + |
| AfrEquat | Gosat | + | ∅ | +++ | +++ |
| 32 | Iasi | ∅ | ∅ | ++ | +++ |
|  | Surf | ∅ | ∅ | + | ++ |
| SouthernAfr | Gosat | ∅ | ∅ | ∅ | ∅ |
| 10 | Iasi | ∅ | ∅ | ∅ | ∅ |
|  | Surf | ∅ | ∅ | ∅ | ∅ |
| Europe | Gosat | + | ∅ | ∅ | ∅ |



Table 3: (continued) Detection of the signal consisting in the anomalies at the 3-yearly time-scale.

| Region | System | 2000-2002 | 2003-2005 | 2006-2008 | 2009-2011 |
|---|---|---|---|---|---|
| 33 | Iasi | + | ∅ | ∅ | ∅ |
| | Surf | + | ∅ | ∅ | ∅ |
| EastEurRussia | Gosat | ∅ | ∅ | ∅ | ∅ |
| 30 | Iasi | ∅ | ∅ | ∅ | ∅ |
| | Surf | ∅ | ∅ | ∅ | ∅ |
| MiddleEast | Gosat | - - | ∅ | + | ++ |
| 16 | Iasi | ∅ | ∅ | ∅ | ∅ |
| | Surf | ∅ | ∅ | ∅ | ∅ |
| SiberianLowlands | Gosat | ∅ | ∅ | ∅ | ∅ |
| 8 | Iasi | ∅ | ∅ | ∅ | ∅ |
| | Surf | ∅ | ∅ | ∅ | ∅ |
| SiberianHighlands | Gosat | + | ∅ | ∅ | ∅ |
| 5 | Iasi | ∅ | ∅ | ∅ | ∅ |
| | Surf | + | ∅ | ∅ | ∅ |
| FarEastSib | Gosat | ++ | + | ∅ | + |
| 1 | Iasi | + | + | ∅ | ∅ |
| | Surf | ++ | + | ∅ | + |
| CentralAsia | Gosat | - | ∅ | ∅ | ∅ |
| 28 | Iasi | ∅ | ∅ | ∅ | ∅ |
| | Surf | ∅ | ∅ | ∅ | ∅ |
| India | Gosat | ∅ | ∅ | ∅ | ∅ |
| 50 | Iasi | ∅ | ∅ | ∅ | ∅ |
| | Surf | ∅ | ∅ | ∅ | ∅ |
| China | Gosat | - | ∅ | ∅ | ∅ |
| 64 | Iasi | - | ∅ | + | ∅ |
| | Surf | - | ∅ | + | + |
| Indonesia | Gosat | + | ∅ | + | ∅ |
| 36 | Iasi | ++ | ∅ | ++ | ∅ |





Table 3: (continued) Detection of the signal consisting in the anomalies at the 3-yearly time-scale.

| Region | System | 2000-2002 | 2003-2005 | 2006-2008 | 2009-2011 |
|---|---|---|---|---|---|
| | Surf | ∅ | ∅ | ∅ | ∅ |
| AustrNZ | Gosat | + | ∅ | ∅ | ∅ |
| 6 | Iasi | + | ∅ | ∅ | ∅ |
| | Surf | + | ∅ | ∅ | ∅ |





## Appendix A: Supplementary tables

Table 4: Yearly mean number of observations over the period used for the Monte-Carlo noise computation (10/2009-09/2010) in the various regions for the three observing systems.

| Region | Area (x$10^6$km$^2$) | GOSAT | IASI | SURF |
|---|---|---|---|---|
| Global | 510 | 32348 | 240084 | 1722 |
| BorN | 31 | 92 | 00 | 172 |
| MidN | 91 | 9060 | 00 | 556 |
| TropN | 126 | 14934 | 121756 | 602 |
| TropS | 128 | 6118 | 107148 | 156 |
| MidS | 95 | 2132 | 9078 | 140 |
| BorS | 37 | 00 | 00 | 96 |
| NorthAmBor | 14 | 194 | 00 | 00 |
| USA | 11 | 2516 | 2218 | 124 |
| CentralAm | 05 | 608 | 6328 | 24 |
| Amazonia | 07 | 802 | 3366 | 00 |
| SouthSAm | 10 | 1780 | 3068 | 24 |
| NorthAfrWest | 10 | 4986 | 4564 | 94 |
| NorthAfrEast | 07 | 3756 | 5148 | 00 |
| AfrEquat | 07 | 1394 | 3572 | 14 |
| SouthernAfr | 07 | 1488 | 3246 | 28 |
| Europe | 06 | 572 | 00 | 94 |
| EastEurRussia | 07 | 896 | 00 | 00 |
| MiddleEast | 06 | 2456 | 3748 | 26 |
| SiberianLowlands | 02 | 170 | 00 | 00 |




Table 4: (continued) Yearly mean number of observations.

| Region | Area (x$10^6$km$^2$) | GOSAT | IASI | SURF |
|---|---|---|---|---|
| SiberianHighlands | 05 | 126 | 00 | 00 |
| FarEastSib | 03 | 54 | 00 | 00 |
| CentralAsia | 12 | 3864 | 694 | 74 |
| India | 03 | 1180 | 4190 | 00 |
| China | 05 | 1164 | 4574 | 00 |
| Indonesia | 07 | 312 | 3324 | 26 |
| AustrNZ | 10 | 3308 | 4362 | 50 |



Table 5: Detection of the signal consisting in the anomalies at the "seasonal" time-scale (JFM, AMJ, JAS, OND). The signal is the difference between each quarter in the 2000-2011 period (i.e. 48 occurrences) and the 2004-2005 average from PBSURF. The noise is computed at the quarter time-scale from each of the three observation systems, GOSAT, IASI and SURF. See Section 2.4 and Section 2.3 for details. In each cell of the Table, we show X% [±TT] (±YY/±ZZ) where X% is the percentage of quarterly anomalies detected, [±TT] is the difference with REFSURF (Table 1), ±YY is the difference in the number of positive anomalies detected compared to REFSURF and ±ZZ is the difference in the total number of detected anomalies compared to REFSURF. Ave. mass= average emitted mass of $CH_4$ over 2004-2005.

| Region<br>Ave. mass (Tg)<br>REFSURF/PBSURF | Gosat | Iasi | Surf |
|---|---|---|---|
| Global 517/499 | 93 [-4] (-11/-2) | 75 [-18] (-10/-9) | 85 [-8] (-10/-4) |
| BorN 18/17 | 87 [+37] (0/+18) | 81 [+31] (0/+15) | 87 [+35] (-1/+17) |
| MidN 177/172 | 83 [-4] (-6/-2) | 50 [-4] (0/-2) | 77 [-4] (-4/-2) |
| TropN 194/165 | 64 [-27] (-12/-13) | 37 [-42] (-8/-20) | 39 [-42] (-9/-20) |
| TropS 115/120 | 43 [+18] (+7/+9) | 43 [+21] (+7/+10) | 27 [+17] (+7/+8) |
| MidS 12/25 | 10 [+10] (+2/+5) | 10 [+10] (+2/+5) | 14 [+14] (+3/+7) |
| BorS 1/0 | 97 [+97] (+23/+47) | 10 [+10] (0/0) | 91 [+91] (+20/+44) |
| NorthAmBor 20/8 | 54 [-43] (-11/-21) | 27 [-25] (-1/-12) | 39 [-48] (-6/-23) |
| USA 37/54 | 58 [+38] (+4/+18) | 14 [+14] (+3/+7) | 08 [+8] (+2/+4) |
| CentralAm 17/13 | 12 [+12] (+6/+6) | 25 [+25] (+11/+12) | 10 [+10] (+5/+5) |
| Amazonia 38/31 | 47 [+39] (+17/+19) | 35 [+33] (+14/+16) | 08 [+8] (+3/+4) |
| SouthSAm 30/45 | 47 [+41] (+12/+20) | 25 [+25] (+5/+12) | 25 [+25] (+5/+12) |
| NorthAfrWest 13/13 | 58 [+42] (+4/+20) | 25 [+25] (+12/+12) | 25 [+21] (+10/+10) |
| NorthAfrEast 11/12 | 97 [+77] (+2/+37) | 29 [+23] (+9/+11) | 25 [+23] (+11/+11) |
| AfrEquat 32/33 | 25 [-10] (-14/-5) | 14 [-6] (-8/-3) | 00 [-4] (-2/-2) |
| SouthernAfr 10/14 | 52 [+48] (+10/+23) | 37 [+37] (+9/+18) | 25 [+25] (+7/+12) |
| Europe 33/33 | 31 [+17] (-7/+8) | 08 [+4] (-2/+2) | 37 [+23] (-7/+11) |





Table 5: (continued) Detection of the signal consisting in the anomalies at the "seasonal" time-scale.

| Region<br>Ave. mass (Tg)<br>REFSURF/PBSURF | Gosat | Iasi | Surf |
|---|---|---|---|
| EastEurRussia 30/27 | 47 [+2] (-1/+1) | 04 [-4] (-2/-2) | 18 [-4] (-3/-2) |
| MiddleEast 16/14 | 00 [-14] (-4/-7) | 00 [0] (0/0) | 00 [0] (0/0) |
| SiberianLowlands 8/14 | 97 [+50] (0/+24) | 64 [+52] (+6/+25) | 91 [+56] (0/+27) |
| SiberianHighlands 5/4 | 25 [+3] (0/+1) | 22 [+16] (+8/+8) | 25 [+3] (0/+1) |
| FarEastSib 1/2 | 87 [+71] (+4/+34) | 72 [+56] (+4/+27) | 83 [+67] (+4/+32) |
| CentralAsia 28/32 | 37 [+4] (+3/+2) | 02 [0] (0/0) | 31 [+13] (+5/+6) |
| India 50/45 | 22 [-40] (-7/-19) | 02 [-48] (-11/-23) | 00 [-35] (-12/-17) |
| China 64/46 | 00 [-43] (-11/-21) | 00 [-10] (-3/-5) | 00 [-31] (-9/-15) |
| Indonesia 36/33 | 20 [+14] (+4/+7) | 31 [+15] (+5/+7) | 08 [+2] (+1/+1) |
| AustrNZ 6/6 | 06 [+4] (+1/+2) | 00 [0] (0/0) | 06 [+4] (+1/+2) |





Table 6: Detection of the signal consisting in the anomalies at the yearly time-scale. The signal is the difference between each year in the 2000-2011 period (i.e. 12 occurrences) and the 2004-2005 average from PBSURF. The noise is computed at the yearly time-scale from each of the three observation systems, GOSAT, IASI and SURF. See Section 2.4 and Section 2.3 for details. In each cell of the Table, we show X% [±TT] (±YY/±ZZ) where X% is the percentage of yearly anomalies detected, [±TT] is the difference with REFSURF (Table 2), ±YY is the difference in the number of positive anomalies detected compared to REFSURF and ±ZZ is the difference in the total number of detected anomalies compared to REFSURF. Ave. mass= average emitted mass of $CH_4$ over 2004-2005.

| Region<br>Ave. mass (Tg)<br>REFSURF/PBSURF | Gosat | Iasi | Surf |
|---|---|---|---|
| Global 517/499 | 75 [-8] (0/-1) | 66 [-9] (0/-1) | 50 [-8] (-1/-1) |
| BorN 18/17 | 41 [+33] (+4/+4) | 00 [0] (0/0) | 41 [+33] (+4/+4) |
| MidN 177/172 | 25 [-41] (-6/-5) | 00 [0] (0/0) | 00 [0] (0/0) |
| TropN 194/165 | 16 [-42] (-4/-5) | 00 [-41] (-5/-5) | 08 [-42] (-5/-5) |
| TropS 115/120 | 50 [+25] (+3/+3) | 66 [+33] (+3/+4) | 41 [+25] (+3/+3) |
| MidS 12/25 | 08 [+8] (+1/+1) | 33 [+33] (+1/+4) | 00 [0] (0/0) |
| BorS 1/0 | 00 [0] (0/0) | 00 [0] (0/0) | 00 [0] (0/0) |
| NorthAmBor 20/8 | 41 [+41] (+5/+5) | 08 [+8] (+1/+1) | 16 [+16] (+2/+2) |
| USA 37/54 | 16 [+16] (0/+2) | 16 [+16] (0/+2) | 00 [0] (0/0) |
| CentralAm 17/13 | 00 [0] (0/0) | 25 [+25] (+3/+3) | 00 [0] (0/0) |
| Amazonia 38/31 | 50 [+50] (+6/+6) | 58 [+58] (+7/+7) | 08 [+8] (+1/+1) |
| SouthSAm 30/45 | 33 [+25] (+2/+3) | 25 [+25] (+2/+3) | 25 [+25] (+2/+3) |
| NorthAfrWest 13/13 | 00 [-41] (-5/-5) | 00 [-33] (-4/-4) | 00 [-41] (-5/-5) |
| NorthAfrEast 11/12 | 00 [-50] (-6/-6) | 00 [-25] (-3/-3) | 00 [-8] (-1/-1) |
| AfrEquat 32/33 | 08 [-33] (-5/-4) | 00 [-33] (-4/-4) | 00 [-33] (-4/-4) |
| SouthernAfr 10/14 | 25 [+25] (+2/+3) | 16 [+16] (+1/+2) | 00 [0] (0/0) |
| Europe 33/33 | 50 [+34] (-2/4) | 41 [+33] (-1/4) | 50 [+34] (-2/4) |





Table 6: (continued) Detection of the signal consisting in the anomalies at the yearly time-scale.

| Region<br>Ave. mass (Tg)<br>REFSURF/PBSURF | Gosat | Iasi | Surf |
|---|---|---|---|
| EastEurRussia 30/27 | 16 [+16] (+1/+2) | 00 [0] (0/0) | 00 [0] (0/0) |
| MiddleEast 16/14 | 00 [-58] (-4/-7) | 00 [0] (0/0) | 00 [0] (0/0) |
| SiberianLowlands 8/14 | 25 [+25] (+2/+3) | 00 [0] (0/0) | 16 [+16] (+1/+2) |
| SiberianHighlands 5/4 | 00 [-8] (-1/-1) | 00 [-8] (-1/-1) | 00 [-8] (-1/-1) |
| FarEastSib 1/2 | 08 [-17] (-2/-2) | 00 [-25] (-3/-3) | 08 [-17] (-2/-2) |
| CentralAsia 28/32 | 00 [-8] (0/-1) | 00 [0] (0/0) | 00 [0] (0/0) |
| India 50/45 | 08 [+8] (0/+1) | 08 [+8] (0/+1) | 00 [0] (0/0) |
| China 64/46 | 00 [0] (0/0) | 00 [0] (0/0) | 00 [-16] (0/-2) |
| Indonesia 36/33 | 50 [+34] (+3/+4) | 66 [+41] (+4/+5) | 25 [+9] (+1/+1) |
| AustrNZ 6/6 | 16 [0] (-1/0) | 08 [+8] (0/+1) | 08 [+8] (0/+1) |