# Peer review of "Can we detect regional methane anomalies? A comparison between three observing systems."

_Atmospheric Chemistry and Physics, 2016_

## Short Comment (SC1) · 4 Apr 2016

I think the main issue with this paper is the fact that it does not clarify well enough the concepts of signal and, in particular, noise. As such it is not clear why is SURFREF is the signal? If you are defining the fluxes from SURFREF as the signal does this mean this is a pseudo-data experiment, or if it is a real data experiment, isn't SURFREF expected to have the same short comings as the SURF inversion? Please also expand on how noise is defined, and why GOSAT has less noise than SURF and these two have less noise than IASI?

Another issue is the fact that too few details are provided in the method section. Please consider expanding on the following issues: - The driving meteorology is nudged to what? - Why is only OH loss considered and not the stratosphere, soil and Cl in the

marine boundary layer? - Is the uncertainty in MCF emissions considered? - The fact that surface observations are not used in the inversions with GOSAT and IASI should be made clear earlier in the paper. - Are there also no long-term trends in the anthropogenic emissions? - Review spatial and temporal correlations assumed in the prior - PBSURF inversion should be introduced earlier and the differences with respect to REFSURF made clearer. - Treatment of input data (e.g. discarding non-background conditions, treatment of flask pairs, more on the model data mismatch). - Please expand a bit more on how the monte carlo ensemble works and explain if you calculate fluxes or only error statistics.

In general I found the way much of the results were given in tables quite difficult to understand particularly for the regional spatial scale. This could be substituted in the following ways: - Maps for each observation system for each temporal scale showing the detection rates at regional spatial scale. - A map for each the observation system showing the time scale at which best detection rates were found. - I think it would be a great contribution to provide maps that delineate the regions of spatial agregation that provide the best detection rates for chosen observing system - Finally, with respect to the seasonal time scale, I think it would be useful if you could provide a plot seasonal cycle (month vs flux) estimated with each of the observing systems for each spatial scale as well as an estimate on how much the OH is contributing to the seasonal cycle?

Finally, please expand more at the section were you compare with Bergamaschi 2013.
* * *

---

## Author Comment (AC1) · 19 Apr 2016

We thank T. G. Nuñez Ramirez for his comments. We answer them in the following. The comments are in bold and our answers in normal font.

**I think the main issue with this paper is the fact that it does not clarify well enough the concepts of signal and, in particular, noise. As such it is not clear why is SURFREF is the signal?**
We chose REFSURF as the signal since it is the inversion that covers the longest time period. Therefore, as explained in Section 2.4, we assume that the inter-annual variability of the inverted fluxes can be more robustly computed over this period (2000-2011) than over the only 2 or 3 years available with the other inversions (e.g. satellite based).

[Figure]

**If you are defining the fluxes from SURFREF as the signal does this mean this is a pseudo-data experiment, or if it is a real data experiment, isn't SURFREF expected to have the same short comings as the SURF inversion?**

REFSURF is a real data experiment since we use the actual real data in this inversion. SURF refers to an ensemble of inversions from which posterior (Bayesian) error statistics are computed. Therefore, the information we get from SURF (posterior flux errors referred to as noise) is not the same as what we get from REFSURF (fluxes themselves).

**Please also expand on how noise is defined, and why GOSAT has less noise than SURF and these two have less noise than IASI?**

We agree that clarifications are needed. The noise is the posterior error variance computed for the IAV of the posterior fluxes for the ensemble of inversions. Indeed, for each dataset (surface observations = SURF, GOSAT and IASI), we perform 10 inversions of 1 year, varying the inversion setup according to an objective analysis described in Cressot *et al.* (2014). The standard deviation of each ensemble allows computing an estimation of the residual error on methane fluxes which we call "noise". We get three "noises", one for each dataset. We will clarify these explanations in the revised version of the manuscript.

The differences in the noises are mainly due to the constraints that each observing system brings on the fluxes. This is linked to the number of data, to their distribution in time and space, and also to their sensitivity to methane fluxes (whether they "see" actual surface fluxes) and to their uncertainty. The noise depends on the region but very often GOSAT has a smaller noise than SURF and IASI because it has more data and they are also more sensitive to the surface (GOSAT "sees" the boundary layer) than IASI data (IASI "sees" the free troposphere only). SURF sometimes does better than IASI because the stations are mostly in the boundary layer (and therefore "see" the surface fluxes directly) whereas IASI provides more integrated information (as it "sees" the free troposphere only).

**Another issue is the fact that too few details are provided in the method section. Please consider expanding on the following issues: - The driving meteorology is nudged to what?**

We forgot to clarify this: we use ECMWF analysed winds. Again, the method section will be expanded also considering the previous comment.

**- Why is only OH loss considered and not the stratosphere, soil and Cl in the marine boundary layer?**

The stratosphere is considered through O1D loss but also OH loss which applies both in the troposphere and the stratosphere. Soil is not considered as such as inversions infer net surface emissions, including the soil uptake. The Cl loss in the marine boundary layer is not implemented yet in our model. This is a limitation of the model that will be acknowledged in the revised version; the implementation of the required reaction is currently in development.

**Is the uncertainty in MCF emissions considered?**

Yes, it is considered: it is taken into account in the **B** matrix, as for methane emissions, and is set at 1% (MCF emissions are fairly well-known and therefore allow constraining OH concentrations effectively, at least until these emissions became negligible).

**- The fact that surface observations are not used in the inversions with GOSAT and IASI should be made clear earlier in the paper.**

We will do this in Section 2.1.

**- Are there also no long-term trends in the anthropogenic emissions?**

Only net emissions are inferred and REFSURF includes the regional trend of anthropogenic emissions.

**- Review spatial and temporal correlations assumed in the prior**

As stated in Cressot *at al.* (2014): "spatial correlations are defined by an e-folding length of 500 km over land and 1000 km over the ocean, without correlation between land and ocean. Temporal correlations are defined by an e-folding length of 2 weeks".

It was checked that combining all errors (variances and covariances from the correlations) leads to a budget uncertainty which is consistent with that of current bottom-up inventories as described in Kirschke *et al.* (2013). This point can be precised in the revised version.

**- PBSURF inversion should be introduced earlier and the differences with respect to REFSURF made clearer.**
We introduce PBSURF only later so as not to confuse the reader with the different inversions defining signal and noises. The main differences between REFSURF and PBSURF are:

- PBSURF uses an analytical inversion whereas REFSURF is variationnal

- because of this, PBSURF uses big regions whereas REFSURF works at the pixel scale

- as a consequence, the **B** matrices of the two inversions are quite different

- PBSURF uses monthly means of the surface observations as constraints whereas REFSURF uses hourly data

- PBSURF retrieves monthly fluxes whereas REFSURF retrieves fluxes at a weekly resolution.

**- Treatment of input data (e.g. discarding non-background conditions, treatment of flask pairs, more on the model data mismatch).**
More on this topics is available in Cressot *et al.* (2014). We will make a more extensive summary of Cressot *et al.* (2014) but details can be found in this former paper.

**- Please expand a bit more on how the monte carlo ensemble works and explain if you calculate fluxes or only error statistics.**

[Figure]

We have generated an ensemble of fluxes from ensembles of inversions, that allowed us to compute Bayesian error statistics. We will expand the description.

**In general I found the way much of the results were given in tables quite difficult to understand particularly for the regional spatial scale. This could be substituted in the following ways: - Maps for each observation system for each temporal scale showing the detection rates at regional spatial scale.**
For the revised version, we can propose the attached maps which correspond to the detection rates given in Tables 1, 2, 5 and 6.

**- A map for each the observation system showing the time scale at which best detection rates were found.**
We can propose the attached synthetic maps but we are not fully convinced that they are very useful compared to the tables.

**- I think it would be a great contribution to provide maps that delineate the regions of spatial agregation that provide the best detection rates for chosen observing system**
This would indeed be interesting but it goes further than the aim of this paper. Finding associations of pixels which optimize the signal-to-noise ratio would be very costly, as explained in Section 3.2.3 of Berchet *et al.* (2015, GMD).

**- Finally, with respect to the seasonal time scale, I think it would be useful if you could provide a plot seasonal cycle (month vs flux) estimated with each of the observing systems for each spatial scale as well as an estimate on how much the OH is contributing to the seasonal cycle?**
The paper is more focused on a signal-to-noise ratio analysis for IAV than a detailed analysis of the seasonality of methane fluxes. We think the paper already contains a lot of material and adding seasonal analysis is another angle than the one we chose here.

**Finally, please expand more at the section were you compare with Bergamaschi**

**2013.**
We will expand the discussion in the revised version.

[Figure]

**Fig. 1.** Detection rate (%) of the signal consisting in the anomalies at the "seasonal" time-scale i.e. quarters of the year, as in Table 1, for GOSAT.

[Figure]

**Fig. 2.** Detection rate (%) of the signal consisting in the anomalies at the "seasonal" time-scale i.e. quarters of the year, as in Table 1, for IASI.

[Figure]

**Fig. 3.** Detection rate (%) of the signal consisting in the anomalies at the "seasonal" time-scale i.e. quarters of the year, as in Table 1, for SURF.

[Figure]

**Fig. 4.** Detection rate (%) of the signal consisting in the anomalies at the yearly time-scale, as in Table 2, for GOSAT.

[Figure]

**Fig. 5.** Detection rate (%) of the signal consisting in the anomalies at the yearly time-scale, as in Table 2, for IASI.

[Figure]

**Fig. 6.** Detection rate (%) of the signal consisting in the anomalies at the yearly time-scale, as in Table 2, for SURF.

[Figure]

**Fig. 7.** Detection rate (%) of the signal consisting in the anomalies at the "seasonal" time-scale, as in Table 5, for GOSAT.

[Figure]

**Fig. 8.** Detection rate (%) of the signal consisting in the anomalies at the "seasonal" time-scale, as in Table 5, for IASI.

[Figure]

**Fig. 9.** Detection rate (%) of the signal consisting in the anomalies at the "seasonal" time-scale, as in Table 5, for SURF.

[Figure]

[Figure]

**Fig. 10.** Detection rate (%) of the signal consisting in the anomalies at the yearly time-scale, as in Table 6, for GOSAT.

[Figure]

**Fig. 11.** Detection rate (%) of the signal consisting in the anomalies at the yearly time-scale, as in Table 6, for IASI.

[Figure]

**Fig. 12.** Detection rate (%) of the signal consisting in the anomalies at the yearly time-scale, as in Table 6, for SURF.

**Fig. 13.** Synthetic map showing the time-scale at which the best detection rate is found for GOSAT.

[Figure]

**Fig. 14.** Synthetic map showing the time-scale at which the best detection rate is found for IASI.

[Figure]

**Fig. 15.** Synthetic map showing the time-scale at which the best detection rate is found for SURF.

---

## Referee Comment (RC1) · Anonymous Referee #1 · 25 Apr 2016

This manuscript investigates the spatio-temporal resolution of global methane inversions using different observing systems. This information is useful for a proper interpretation of inversion results obtained using existing observing systems, and for the design of new systems. An expected outcome is that larger regions are better resolved than smaller regions. Less expected is the finding that smaller regions are better resolved at the seasonal than at the inter-annual time scale. While trying to understand this, a couple of questions came up, as explained below, which I found have not been dealt with adequately yet. To make this study acceptable for publication this will have to be repaired, and/or explained more clearly.

GENERAL COMMENTS

Figure 3 and the tables depend on a detection criterion, like threshold SNR value which

should be exceeded to declare a region as detected or not. This criterion should be defined explicitly in the text. From one of the figure captions I found out that the criterion corresponds to SNR=1. This sounds like a rather loose criterion. Wouldn't something like a 95% confidence criterion be more appropriate? Whatever choice is made it should be stated and motivated clearly.

In this study the REFSURF inversion is used as reference, representing what the true variability would be like. As long as we don't know the true flux the results of an inversion may seem a defensible choice. However, this is only true as long as the validity of this approximation doesn't interfere with the conclusions that are derived from it in the end. Since this reference set of fluxes comes from an inversion of surface data itself, it suffers from the same flux detection limitations as the SURF inversion. Suppose that the setups of REFSURF and SURF were statistically equivalent, wouldn't you expect SNR=1? I mean if their posterior uncertainties are the same then REFSURF would be like a random instance of the posterior uncertainty of SURF. In this case what remains is equivalent to a comparison of the posterior uncertainties of the SURF, IASI, and GOSAT inversions.

The comment above has implications for the conclusions regarding the scale dependency of flux detection. For example, if REFSURF is not capable of resolving small-scale variability, it will generate noise (depending on the a priori constraints). If the use of GOSAT leads to better-constrained small-scale fluxes it may end up 'detecting' the noise of the REFSURF inversion, rather than the variability of the true fluxes at that scale. All we learn in the end is that GOSAT is better able to resolve small-scale fluxes than the surface network. That is something we could have concluded already looking only at their posterior uncertainties. Then what is the added value of the method that is used here?

The choice of reference period for calculating the signal should be explained better. It is chosen because 'it corresponds to a period of minimum atmospheric growth rate'. I guess this means that it has a minimal contribution from the long-term trend. However,

shouldn't it be representative of the entire period also? The fact that you get anomalies that are predominantly positive suggests it is not. As a result your IAV signal will inevitably contain signal from a time scale outside the 3-year IAV window. It becomes even worse for the annual and seasonal time scale. I don't see how the method avoids signals from longer time-scales affecting the seasonal time scale. Wouldn't I have been better to take out variations on longer time-scales before computing seasonal anomalies?

The role of the prior flux uncertainty should be explained better. I wonder if some regions get already 'detected' without using any data, just because the prior uncertainty is already small enough to satisfy the detection criterion. It would explain why some regions are detected without the observing contributing any significant constraint (for example IASI detecting fluxes from NorthAmericanBoreal, when only data between 30S and 30N are used).

Another factor influencing the scale dependency of flux detection is the accuracy at which posterior fluxes are approximated. Give that only 10 Mont Carlo ensemble members are used this accuracy cannot be that high (see for example the appendix of Pandey et al, 2016 for a formula to compute the uncertainty of a Monte Carlo derived uncertainty for a given number of iterations). Although the limited ensemble size should not introduce a scale dependency, the number of iterations per inversion could do that, because, depending on the search algorithm used, the large scales may be solved first being the dominant eigenvectors of the optimization problem. No information is given about the number of iterations that is used, but in our experience the M1QN3 could converge slowly. Therefore additional information about the convergence of small-scale fluxes is needed.

Somewhere in the discussion a note of caution is required that the posterior uncertainties are derived without proper accounting for systematic errors in the satellite retrieval and transport model. Because of this, despite the use of real data, the detection statistics probably end up being rather optimistic. The use of Desroziers recipe for error

tuning does not account for the neglect of off diagonals in the R matrix.

SPECIFIC COMMENTS

Page 3, line 5: How appropriate is it here to use prior fluxes without IAV? It means that the prior is biased with regard to IAV, and as a result the posterior IAV will be low biased too (assuming that all other statistical assumptions are satisfied).

Page 4, line 1: I'm trying to understand the logic of the sqrt(2). How do you define inter-annual variability? Wouldn't it be the difference from one year to another? A sqrt(2) inflation rather suggests the variability of the 2-yearly flux in Tg/(2 year). How does that fit with the 3 year time windows? Apart from this I don't see why the assumption of uncorrelated errors would lead to a conservative estimate. I would rather think of posterior uncertainties as being negatively correlated because of limitations in inde-pendently resolving the yearly fluxes. Because of these complications I wonder how appropriate it is to address the 3 yearly time scale using a one year inversion anyway.

Page 4, line 6: 'The uncertainty in OH (5% after optimization)' Which scale does this refer to? If it is the global scale, then how about the uncertainty per latitude band?

Page 9, line 7: 'PBSURF signal is twice as often detected' Why is this? I guess it depends on the size of the regions in PBSURF compared to the scale of the regions that are evaluated. If the latter are smaller then wouldn't you rather expect that the large-region inversion suppresses the within-region variability? In that case they would become harder, rather than easier to detect. Some further discussion at this point would be helpful.

TECHNICAL COMMENTS

Page 9, line 7: "REFSURF" i.o. "SURF". Please check if there are other instances where REFSURF was meant.

Page 11, line 11: 'acknowledge' i.o. 'aknowledge'

[Figure]

Figure 5: What happens to the whisker-boxes at the 2Y time scale? I guess they become too compressed to see. If so, then please mention this somewhere (it shows up in other figures also).

Figure 5, 6 & 7, caption: 'aggregation' i.o. 'agregation'

---

## Referee Comment (RC2) · Anonymous Referee #2 · 27 Apr 2016

This paper shows results from a set of CH4 inversions using three different observation sets (in situ, IASI, and GOSAT) to test whether anomalies in flux can be detected across a range of time and spatial scales. The ultimate goal is to determine whether such inversions can be used to attribute methane flux signals, like the change in global growth rate through the 2000s, to a particular region or regions and, perhaps, biogeochemical processes. The authors have done a lot of work to make the results statistically meaningful, the approach is generally sound, figures and tables are informative, and the discussion is accurate, if perhaps not fully satisfying. In my opinion the material is clearly worthy of publication in ACP after satisfying the concerns of the reviewers.

The paper suffers at times from lack of clarity and some inverse methodological issues exist, which are well-characterized by Anonymous Reviewer #1 and the comment from T. G. Nuñez Ramirez. I did not find the tables too difficult, but they do take some focus.

These issues aside, the question that remains to me is: so what? What are the implications of the findings for using CH4 measurements and inverse models to understand the underlying processes? What is the message for carbon cycle science? To my reading the answer to the title question is: NO, except on the broadest of scales and strongest of signals (seasonality), which doesn't really require very extensive measurements or sophisticated mathematical techniques and holds little useful information. This is a serious problem for understanding the current CH4 budget, for projecting future interactions of CH4 and climate, and for designing mitigation policies to reduce the radiative forcing of CH4. The paper alludes to some of the most egregious shortcomings, but never really comes out and says our data and techniques are inadequate and what should be done about it. I fully agree with other comments that setting the detection criterion to SNR =1 is a very low bar for attributing anomalies to specific locations and processes. The paper is also sometimes seemingly overly optimistic about the model ability to capture signals, e.g., Conclusions line 8-11, where having any detectable anomalies ($\sim$25% on average) is called 'fair to good' and Abstract, where regional scale signals are said to be 'properly detected.' Clearly something much better than current observations and/or existing model formulations is needed. I think the paper should not shy away from such a statement and point out specifically where the problems reside in the analysis. The fact that the detectability depends on the underlying (modeled) signal configuration is further indictment of the overall flux analysis method. The statements that inversions 'should always include an uncertainty assessment', 'attribution. . . needs more attention', and 'more observations and . . . improved transport' are platitudes that don't require a detailed analysis like the one produced in this paper. Go ahead and give the discussion some punch.

Minor Comments: The analysis does not address transport issues at all, although perhaps it could. Such analysis could include impact of transport uncertainty on inference of fluxes in unobserved regions (e.g., satellite data in dark or high latitudes) and resulting 'noise.' Expand discussion or delete from Conclusions lines 32-33.

Not clear that detection of anomalies at grid scale in Amazon is robust. Depends on signal, which may not be realistic from sparse data constraint. Maybe examine more closely or moderate expectations.
* * *

---

## Author Response (AR1)

**Answers to reviewer # 1**

We thank the reviewer for his/her comments. We answer them in the following. The comments are in bold and our answers in normal font, the new text being in blue.

**This manuscript investigates the spatio-temporal resolution of global methane inversions using different observing systems. This information is useful for a proper interpretation of inversion results obtained using existing observing systems, and for the design of new systems. An expected outcome is that larger regions are better resolved than smaller regions. Less expected is the finding that smaller regions are better resolved at the seasonal than at the inter-annual time scale. While trying to understand this, a couple of questions came up, as explained below, which I found have not been dealt with adequately yet. To make this study acceptable for publication this will have to be repaired, and/or explained more clearly.**

**GENERAL COMMENTS**

**Figure 3 and the tables depend on a detection criterion, like threshold SNR value which should be exceeded to declare a region as detected or not. This criterion should be defined explicitly in the text. From one of the figure captions I found out that the criterion corresponds to SNR=1. This sounds like a rather loose criterion. Wouldn't something like a 95% confidence criterion be more appropriate? Whatever choice is made it should be stated and motivated clearly.**

We have modified Section 2.4 and added the missing information:

"Our criterion consists in evaluating the ability of the observing systems to detect $CH_4$ anomalies of a given amplitude, defined by the reference inversion. For this, we define a signal-to-noise ratio:

- the inversion with surface measurements is chosen to provide the signal as the data covers a long time window (2000-2011) as compared to the two other observing systems. This longer window makes it possible to sample the $CH_4$ IAV more robustly than a 2-3 year inversion. We assume that the fluxes inferred by this inversion are representatitve of state-of-the art inversions currently published. The signal is actually the $CH_4$ anomalies for the various time scales derived from REFSURF.

- for the three observing systems (SURF, IASI and GOSAT), the Bayesian posterior errors of the year-to-year changes of $CH_4$ fluxes, computed

Finally, the criterion for detecting $CH_4$ anomalies is that the signal-to-noise ratio is larger than 1 ($\approx$68% confidence). "

It would be more robust to use SNR=2 i.e. 95% but then almost nothing would be considered as detected so that would prevent any constructive comments. We also have stressed this point in the conclusion "Our criterion is based on a 68% confidence interval (1 sigma). At almost all regional time-space scales (except in NorthAmbor, AfrEquat at the longer time-scales and a few cases in India, Indonesia, EastEurRussia and FarEastSib), the three observing systems would fail the test at 2 sigmas (95%), a more stringent criterion commonly used in other scientific communities."

**In this study the REFSURF inversion is used as reference, representing what the true variability would be like. As long as we don't know the true flux the results of an inversion may seem a defensible choice. However, this is only true as long as the validity of this approximation doesn't interfere with the conclusions that are derived from it in the end. Since this reference set of fluxes comes from an inversion of surface data itself, it suffers from the same flux detection limitations as the SURF inversion. Suppose that the setups of REFSURF and SURF were statistically equivalent, wouldn't you expect SNR=1? I mean if their posterior uncertainties are the same then REFSURF would be like a random instance of the posterior uncertainty of SURF. In this case what remains is equivalent to a comparison of the posterior uncertainties of the SURF, IASI, and GOSAT inversions.**

**The comment above has implications for the conclusions regarding the scale dependency of flux detection. For example, if REFSURF is not capable of resolving small-scale variability, it will generate noise (depending on the a priori constraints). If the use of GOSAT leads to better-constrained small-scale fluxes it may end up 'detecting' the noise of the REFSURF inversion, rather than the variability of the true fluxes at that scale. All we learn in the end is that GOSAT is better able to resolve small-scale fluxes than the surface network. That is something we could have concluded already looking only at their posterior uncertainties. Then what is the added value of the method that is used here?**

REFSURF and SURF couldn't be statistically equivalent: REFSURF is driven by the observation vector whereas SURF is driven by the statistics

in the covariance matrices (of observations and the state vector) and by the (transport-)model.

In this paper we assume that REFSURF reasonably represents the discussed scale as other state-of-the art inversions. We agree with the reviewer's comment on the implications if this assumption was wrong, and we tried to clarify even more this assumption in the text e.g. in Section 2.4 "We assume that the fluxes inferred by this inversion are representatitve of state-of-the art inversions currently published." and "However, the quality of the chosen signal remains debatable and our diagnostic for GOSAT and IASI may be pessimistic in areas where SURF signal-to-noise ratio is low." We could have compared the noises of the three systems. But positionning the noises with the signal allows to go further than comparing the three observing systems. Indeed, the largest noise may nevertheless be small enough compared to the signal we want to detect (or the smallest one be too large). This is why the signal computed from the 12-year REFSURF inversion is used to normalize the noises. The aim is to state whether the inter-annual variability at various time and spatial scales can be detected by one or several of the observing system. This is now stated explicitly in Section 2.4:

"Comparing signal-to-noise ratios amounts to comparing noises normalized by the expected signals. The normalization provides an absolute criterion to assess the time scales and regions at which the $CH_4$ anomalies are reliable."

**The choice of reference period for calculating the signal should be explained better. It is chosen because 'it corresponds to a period of minimum atmospheric growth rate'. I guess this means that it has a minimal contribution from the long-term trend. However, shouldn't it be representative of the entire period also? The fact that you get anomalies that are predominantly positive suggests it is not. As a result your IAV signal will inevitably contain signal from a time scale outside the 3-year IAV window. It becomes even worse for the annual and seasonal time scale. I don't see how the method avoids signals from longer time-scales affecting the seasonal time scale. Wouldn't I have been better to take out variations on longer time-scales before computing seasonal anomalies?**

We agree that the way we defined flux anomalies had to be made clearer. Our aim is not to isolate the various frequencies in the signal due to the fluxes (at seasonal, annual, year-to-year scales and long-term trend). Since our noise refers to the year to year changes in methane emissions at various scales, we defined the signal i.e. "anomalies", as the difference between one given season/year/3year-period and a reference (seasonal/yearly/3yearly mean) to

be consistent with the noise definition. Doing so, a given time scale contains contributions from other time scales. The signal is here mostly to give an order of magnitude, at various time scales, of what we want to detect, considering the year-to-year noises defined for the three observing systems.

The choice of 2004-2005 is mostly arbitrary and we agree that, as it corresponds to a minimum in methane growth rate and internannual variations it leads to more positive anomalies for the longer time scales. This is less true for smaller spatial and temporal scales. We have modified Section 2.4: "$CH_4$ regional flux anomalies are defined here as the deviation from a reference of the $CH_4$ inferred fluxes for various time periods, from the monthly to the 3-yearly scale. The reference is the 2004-2005 mean over the same time-period. The aim of this definition is to get the order of magnitude of the year-to-year changes at various time scales. As the 2004-2005 reference corresponds to a period of minimum atmospheric methane growth rate (Dlugokencky et al., 2011), it leads to more positive anomalies for the longer time scales. "

Taking out the longer time scales to get at the amplitude of seasonal cycles would be another study.

**The role of the prior flux uncertainty should be explained better. I wonder if some regions get already 'detected' without using any data, just because the prior uncertainty is already small enough to satisfy the detection criterion. It would explain why some regions are detected without the observing contributing any significant constraint (for example IASI detecting fluxes from NorthAmericanBoreal, when only data between 30S and 30N are used).**

Section 2.3 now states more clearly how the prior uncertainty is used to compute the posterior uncertainty (used as noise):

"The posterior error statistics (the "noise" for our study) are estimated as follows:

- we estimate the ratio of posterior to prior standard deviations of the annual flux errors $r = \frac{\sigma_a}{\sigma_b}$ from the ensemble, a quantity which is more robust than $\sigma_a$ and $\sigma_b$ individually for small ensembles (because some of the underspread affects the prior and the posterior in a similar way); the number of members in the ensemble depends on the time scale e.g. 10 members for the yearly time scale (10 inversions, each one covering 1 year), 120 members for the monthly time scale

- we estimate the posterior standard deviations of the annual flux errors by multiplying $r$ to the known value of $\sigma_b$ i.e. the one implied by our

error covariance matrix (computed from the above assumptions)

- the posterior standard deviations of the pluri-annual flux errors errors for $n$ years is obtained by applying a factor of $\frac{1}{\sqrt{n}}$ to the previous result, assuming that the errors are uncorrelated from one year to the next

- the posterior standard deviations of the difference between fluxes from one year to the next (i.e. the error on the IAV for two consecutive years) is computed by applying an inflation factor of $\sqrt{2}$ to the previous result, still assuming that the errors are uncorrelated from one year to the next. We assume this apporach to be a conservative hypothesis since in reality some of the transport and retrieval errors are recurrent, thereby inducing positive correlations and reducing the inflation factor."

Our prior fluxes have nearly no any interannual variability and are therefore not supposed to detect anomalies in general. However, we checked that our prior error statistics reflect this property by confronting them with the anomalies in the same way as what we present for the posterior. The property is verified for all regions and time scales (i.e. the detection rate is marginal or null) except for NorthAmBor and BorN at the seasonal time scale, where detection rates of 58% and 37%, respectively, are obtained. This result was correctly intuited by the reviewer and may suggest that our prior error statistics in NorthAmBor are underestimated. We have added a note of caution in the main text: "The detection rate is above 50% for the three observing systems in this region (Table 1), but, in contrast to the other regions and to the other time scales, the prior error statistics already lead to detection rates of 58% for the prior. This shows that the Tropical IASI soundings do not add information for this region and at this time scale, as expected. "

**Another factor influencing the scale dependency of flux detection is the accuracy at which posterior fluxes are approximated. Give that only 10 Mont Carlo ensemble members are used this accuracy cannot be that high (see for example the appendix of Pandey et al, 2016 for a formula to compute the uncertainty of a Monte Carlo derived uncertainty for a given number of iterations). Although the limited ensemble size should not introduce a scale dependency, the number of iterations per inversion could do that, because, depending on the search algorithm used, the large scales may be solved first being the dominant eigenvectors of the**

**optimization problem.**

As now stated in Section 2.3, "we estimate the ratio of posterior to prior standard deviations of the annual flux errors $r = \frac{\sigma_a}{\sigma_b}$ from the ensemble, a quantity which is more robust than $\sigma_a$ and $\sigma_b$ individually for small ensembles (because some of the underspread affects the prior and the posterior in a similar way)."

**No information is given about the number of iterations that is used, but in our experience the M1QN3 could converge slowly. Therefore additional information about the convergence of small-scale fluxes is needed.**

The convergence of the inversions with M1QN3 was stopped on the criterion of the ratio final/initial gradient norm: it must be less than 0.01. This information has been added in Section 2.1.

**Somewhere in the discussion a note of caution is required that the posterior uncertainties are derived without proper accounting for systematic errors in the satellite retrieval and transport model. Because of this, despite the use of real data, the detection statistics probably end up being rather optimistic. The use of Desroziers recipe for error tuning does not account for the neglect of off diagonals in the R matrix.**

A "note of caution" has been added in the discussion in Section 5, Conclusions: "We also have neglected the impact of likely state-dependent systematic errors in current satellite retrievals and transport models that further reduce the inversion performance to an unknown extent."

**SPECIFIC COMMENTS**

**Page 3, line 5: How appropriate is it here to use prior fluxes without IAV? It means that the prior is biased with regard to IAV, and as a result the posterior IAV will be low biased too (assuming that all other statistical assumptions are satisfied).**

In inversion, the choice of no prior IAV is generally made to minimize the influence of prior emissions on the inferred signals and let observations generate the IAV part of it. We choose this assumption following the same spirit although we agree it may smooth the generated IAV. Indeed, Bergamaschi *et al.* (2013) tested such "flat" priors (their S3 scenarios) and showed that the IAV was derived from the assimilated data. We added a sentence about this in Section 2.1: "It is important here to recall that the prior fluxes (fires excepted) have no inter-annual variability (IAV). This choice is made for IAV to be generated by atmospheric observations and atmospheric transport and chemistry and not by prior IAVs of emissions (and sinks) which are still uncertain or even controversial (e.g. Schaefer et al. (2016); Hausmann

et al. (2016); Nisbet et al. (2014))"

**Page 4, line 1: I'm trying to understand the logic of the sqrt(2). How do you define inter-annual variability? Wouldn't it be the difference from one year to another? A sqrt(2) inflation rather suggests the variability of the 2-yearly flux in Tg/(2 year). How does that fit with the 3 year time windows? Apart from this I don't see why the assumption of uncorrelated errors would lead to a conservative estimate. I would rather think of posterior uncertainties as being negatively correlated because of limitations in independently resolving the yearly fluxes. Because of these complications I wonder how appropriate it is to address the 3 yearly time scale using a one year inversion anyway.**

The explanation in Section 2.3 and 2.4 have been made clearer; regarding the $\sqrt{2}$ factor: "the posterior standard deviations of the difference between fluxes from one year to the next (i.e. the error on the IAV for two consecutive years) is computed by applying an inflation factor of $\sqrt{2}$ to the previous result, still assuming that the errors are uncorrelated from one year to the next. We assume this approach to be a conservative hypothesis since in reality some of the transport and retrieval errors are recurrent, thereby inducing positive correlations and reducing the inflation factor." One issue with this hypothesis is that the limitation induced by observing systems to solve independently the methane fluxes also leads to negatively correlated errors. Remember that we use standard deviations, so that the standard deviation for a difference is the standard deviation of the sum of the two terms, here one year and the next one; since the two terms are the same (because we have only one year in the ensembles), we get that the standard deviation of the difference from one year to the next is $\sqrt{2} \times$ the standard deviation of the available year.

**Page 4, line 6: 'The uncertainty in OH (5% after optimization)' Which scale does this refer to? If it is the global scale, then how about the uncertainty per latitude band?**

5% refers to the global scale and we did not compute the uncertainty for the different latitudinal bands. However, as prior error are already low for OH, we expect very little changes compared to the prior values.

**Page 9, line 7: 'PBSURF signal is twice as often detected' Why is this? I guess it depends on the size of the regions in PBSURF compared to the scale of the regions that are evaluated. If the latter are smaller then wouldn't you rather expect that the large-region inversion suppresses the within-region variability? In that case they would become harder, rather than easier to detect.**

**Some further discussion at this point would be helpful.**

This is a good point. Large-region based inversions relying on sparse surface networks generally generate larger IAVs than pixel-based inversions when aggregating the results of the latter at the same regional scale. One illustration of this can be found in Pison *et al.*, 2013. This can be understood as a large-region based inversion scales up or down a whole region and changes can be large, especially when the atmospheric constraints are far from the region (e.g. South America in Pison *et al.*, 2013). In pixel based inversions, as we only prescribe loose spatial correlation, changes appear to be more located around the stations. We added this in Section 4.2: "The large-region-scale inversion means that the spatial variability of the prior is kept within each region and is only scaled (contrary to REFSURF, which is performed at the pixel scale i.e. is able to vary only a few pixels to match the data). This difference in the methods may lead to very different spatial variability in each of the regions of interest (Figure 4), a larger variability allowing a better detection rate with our criterion. Indeed, the large-region-scale inversion may lead to larger variability than pixel-based inversions in some regions (e.g. Pison et al., 2013) because of the homothetic scaling of the pixels composing each region in PBSURF (correlations between pixels of 1) as opposed to the individual scaling of model pixels with soft constraints in REFSURF (spatial correlations less than 1))."

**TECHNICAL COMMENTS**

**Page 9, line 7: 'REFSURF' i.o. 'SURF'. Please check if there are other instances where REFSURF was meant.**
OK

**Page 11, line 11: 'acknowledge' i.o. 'aknowledge'**
OK.

**Figure 5: What happens to the whisker-boxes at the 2Y time scale? I guess they become too compressed to see. If so, then please mention this somewhere (it shows up in other figures also).**
There are no whisker-boxes for the 3Y scale since there is only one window of three years in each of the 3-year periods (the whisker-boxes for Y are made with 3 values, 6 values for 6M, etc).

**Figure 5, 6 & 7, caption: 'aggregation' i.o. 'agregation'**
OK.

**Answers to reviewer #2**

We thank the reviewer for his/her comments. We answer them in the following. The comments are in bold and our answers in normal font, the new text being in blue.

**This paper shows results from a set of CH4 inversions using three different observation sets (in situ, IASI, and GOSAT) to test whether anomalies in flux can be detected across a range of time and spatial scales. The ultimate goal is to determine whether such inversions can be used to attribute methane flux signals, like the change in global growth rate through the 2000s, to a particular region or regions and, perhaps, biogeochemical processes. The authors have done a lot of work to make the results statistically meaningful, the approach is generally sound, figures and tables are informative, and the discussion is accurate, if perhaps not fully satisfying. In my opinion the material is clearly worthy of publication in ACP after satisfying the concerns of the reviewers.**

**The paper suffers at times from lack of clarity and some inverse methodological issues exist, which are well-characterized by Anonymous Reviewer #1 and the comment from T. G. Nuñez Ramirez. I did not find the tables too difficult, but they do take some focus.**

We have particularly re-written Sections 2.3 and 2.4 to make them clearer and answered the comments of reviewer #1 and T.G. Nuñez Ramirez.

**These issues aside, the question that remains to me is: so what? What are the implications of the findings for using CH4 measurements and inverse models to understand the underlying processes? What is the message for carbon cycle science? To my reading the answer to the title question is: NO, except on the broadest of scales and strongest of signals (seasonality), which doesn't really require very extensive measurements or sophisticated mathematical techniques and holds little useful information. This is a serious problem for understanding the current CH4 budget, for projecting future interactions of CH4 and climate, and for designing mitigation policies to reduce the radiative forcing of CH4. The paper alludes to some of the most egregious short-comings, but never really comes out and says our data and techniques are inadequate and what should be done about it.**

We modified the end of the paper (Section 5 Conclusions) to acknowledge that the situation of methane atmospheric inversions may not be as optimistic as found in current papers, although our study has limitations that

we also acknowledge more clearly (e.g. underestimation of the signal, over-estimation of the noises).

**I fully agree with other comments that setting the detection criterion to SNR =1 is a very low bar for attributing anomalies to specific locations and processes.**
It would be more robust to use SNR=2 i.e. 95% but then almost nothing would be considered as detected so that would prevent any constructive comments. We also have stressed this point in the conclusion "Our criterion is based on a 68% confidence interval (1 sigma). At almost all regional time-space scales (except in NorthAmbor, AfrEquat at the longer time-scales and a few cases in India, Indonesia, EastEurRussia and FarEastSib), the three observing systems would fail the test at 2 sigmas (95%), a more stringent criterion commonly used in other scientific communities."

**The paper is also sometimes seemingly overly optimistic about the model ability to capture signals, e.g., Conclusions line 8-11, where having any detectable anomalies ($\approx25\%$ on average) is called 'fair to good' and Abstract, where regional scale signals are said to be 'properly detected.'**
Taking into account this comment, we found that we had chosen a too optimistic way of computing the noises. We have therefore corrected the problem and modified the results and the text accordingly.

**Clearly something much better than current observations and/or existing model formulations is needed. I think the paper should not shy away from such a statement and point out specifically where the problems reside in the analysis. The fact that the detectability depends on the underlying (modeled) signal configuration is further indictment of the overall flux analysis method. The statements that inversions 'should always include an uncertainty assessment', 'attribution... needs more attention', and 'more observations and ... improved transport' are platitudes that don't require a detailed analysis like the one produced in this paper. Go ahead and give the discussion some punch.**
We rephrased the end of the paper (Section 5: conclusions) in this sense, although one has to consider that our work has limitations and is not positioned as the ultimate "killer" of atmospheric inversions. We now acknowledge more clearly the challenges for atmospheric inversions and for the work presented here. We still think that the message to deliver systematically an uncertainty assessment in inversion papers is useful to mention as it is not always reported in current papers. Indeed, the cost of running Monte-Carlo ensembles or the explicit computation of posterior errors with the Hessian

matrix or the difficulty in designing relevant sensitivity studies often limit the uncertainty analysis proposed in papers. Our work also shows the critical importance to do so systematically to have the proper level of significance of the inferred fluxes.

**Minor Comments:**

**The analysis does not address transport issues at all, although perhaps it could. Such analysis could include impact of transport uncertainty on inference of fluxes in unobserved regions (e.g., satellite data in dark or high latitudes) and resulting 'noise.' Expand discussion or delete from Conclusions lines 32-33.**

OK, this part has been deleted.

**Not clear that detection of anomalies at grid scale in Amazon is robust. Depends on signal, which may not be realistic from sparse data constraint. Maybe examine more closely or moderate expectations.**

We moderated our statements here.

[revised manuscript text omitted]